# Automatic Synthetic Data and Fine-grained Adaptive Feature Alignment for Composed Person Retrieval

**Delong Liu[1], Haiwen Li[1], Zhaohui Hou[2], Zhicheng Zhao[1,3,4],[*] Fei Su[1,3,4], Yuan Dong[1]**

[1]Beijing University of Posts and Telecommunications
[2]SenseTime
[3]Beijing Key Laboratory of Network System and Network Culture
[4]Key Laboratory of Interactive Technology and Experience System, Ministry of Culture and Tourism
{liudelong, lihaiwen, zhaozc, sufei, yuandong}@bupt.edu.cn
houzhaohui@sensetime.com

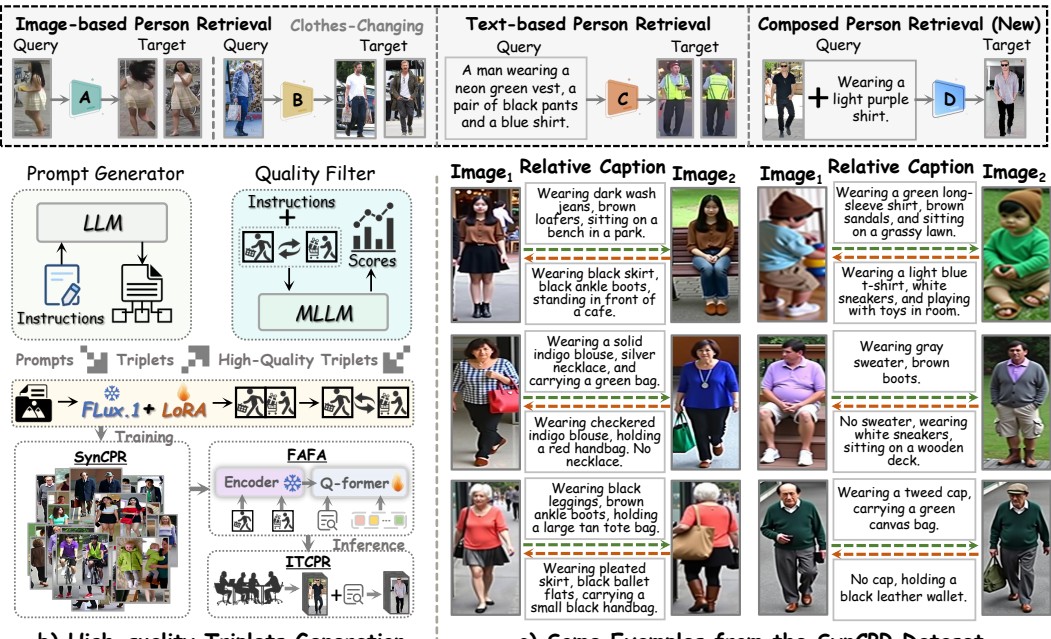

Figure 1: **Overview of our contributions.** (a) Comparison of the proposed composed person retrieval task with several classic person retrieval tasks. (b) Illustration of the proposed automatic high-quality CPR data synthesis pipeline, the proposed training framework FAFA, and the first carefully annotated test set in this domain, ITCPR. (c) Some examples from our fully synthetic SynCPR dataset.

## Abstract

Person retrieval has attracted rising attention. Existing methods are mainly divided into two retrieval modes, namely image-only and text-only. However, they are unable to make full use of the available information and are difficult to meet diverse application requirements. To address the above limitations, we propose a new Composed Person Retrieval (CPR) task, which combines visual and textual queries to identify individuals of interest from large-scale person image databases. Nevertheless, the foremost difficulty of the CPR task is the lack of available annotated datasets. Therefore, we first introduce a scalable automatic data synthesis pipeline,

---

[*]Corresponding author: Zhicheng Zhao.

39th Conference on Neural Information Processing Systems (NeurIPS 2025).

which decomposes complex multimodal data generation into the creation of textual quadruples followed by identity-consistent image synthesis using fine-tuned generative models. Meanwhile, a multimodal filtering method is designed to ensure the resulting SynCPR dataset retains 1.15 million high-quality and fully synthetic triplets. Additionally, to improve the representation of composed person queries, we propose a novel Fine-grained Adaptive Feature Alignment (FAFA) framework through fine-grained dynamic alignment and masked feature reasoning. Moreover, for objective evaluation, we manually annotate the Image-Text Composed Person Retrieval (ITCPR) test set. The extensive experiments demonstrate the effectiveness of the SynCPR dataset and the superiority of the proposed FAFA framework when compared with the state-of-the-art methods. All code and data will be provided at `https://github.com/Delong-liu-bupt/Composed_Person_Retrieval`.

# 1    Introduction

Person retrieval [1, 2] aims to identify target individuals from large-scale databases and encompasses two primary research directions: image-based person retrieval (IPR) [3] and text-based person retrieval (TPR) [4]. Typically, they rely independently on images or textual queries to identify the intended targets. In fact, in real-world scenarios, visual and textual information are often simultaneously available when searching for specific individuals. For example, when looking for a missing person, people may refer to the past photographs along with a recent verbal description. However, existing methods fail to fully exploit this combined information, resulting in suboptimal retrieval accuracy.

To address this drawback, as shown in Figure 1(a), a novel task named Composed Person Retrieval (CPR) is introduced, which fuses visual and textual information for person retrieval. Similar to Composed Image Retrieval (CIR) [5, 6], the CPR data will also comprise numerous triplets ($I_q$, $T_q$, $I_t$), where each triplet consists of a reference person image ($I_q$), a relative caption ($T_q$), and one or more target images4 ($I_t$). The objective is to effectively locate $I_t$ by exploiting the complementary information between $I_q$ and $T_q$. Constructing such data requires paired images of individuals with the same identity (ID) and textual descriptions highlighting their differences. However, manual collection and annotation is time-consuming, costly, and often hindered by privacy issues, limiting both the variety and scale of the depicted scenarios. Consequently, this poses a significant challenge to the construction of a comprehensive, high-quality, large-scale training dataset for CPR task.

To cope with these challenges, we propose a scalable automatic CPR data synthesis pipeline, depicted in Figure 1(b). The generation of complex multimodal triplets is achieved by overcoming two key problems: First, how to create pure and diverse textual data. Second, how to leverage the generative models to transform a subset of this text into identity-consistent person images, thus attaining CPR data synthesis. Specifically, this pipeline is decomposed into three stages. First, a Large Language Model (LLM) [7] generates abundant textual quadruples, and each one comprises two image descriptions and two relative captions that connect them. Through carefully designed prompts, the LLM is guided to produce diverse descriptions reflecting a wide range of individuals and states, while effectively capturing relative differences.

In order to solve the second problem, we first fulfill the synthesis of person image-text pairs in the second stage. Considering that directly employing pretrained diffusion models to individually generate $I_q$ and $I_t$ will lead to identity mismatches and discrepancies from real-world distributions. Thus, we fine-tune generative models [8, 9] using real-world data [4] to derive a suitable person image generator firstly. Subsequently, by merging textual prompts, we simultaneously generate a single image containing two related sub-images, which are then cropped into the reference image and the target one, thereby ensuring identity consistency.

In the third stage, rigorous data filtering method is designed to ensure the high quality of triplets. Specifically, a multimodal large language model (MLLM) [10] is applied to evaluates the generated triplets according to four scoring criteria: image quality, identity consistency, text-image alignment, and relative caption quality. After filtering based on these scores, a high-quality, fully synthetic CPR dataset named Synthetic Composed Person Retrieval (SynCPR) can be obtained, and its representative examples are shown in Figure 1(c).

Moreover, we propose a novel framework tailored for CPR task: Fine-grained Adaptive Feature Alignment (FAFA). FAFA strengthens model training by integrating fine-grained dynamic alignment

with bidirectional masked feature reasoning, thereby generating more comprehensive, robust, and fine-grained representations. Finally, in order to conduct an objective evaluation of FAFA's performance, an Image-Text Composed Person Retrieval (ITCPR) test set is carefully constructed and manually annotated, based on widely-used clothes-changing person retrieval datasets such as Celeb-reID [11], LAST [12], and PRCC [13]. Among them, we annotate the relative captions by selecting images of the same identity in different outfits or states, and ultimately form complete triplets. Extensive experiments on ITCPR dataset demonstrate the effectiveness of both the proposed automated triplet synthesis pipeline and the FAFA framework. The main contributions can be summarized as follows:

- A novel cross-modal task, composed person retrieval is proposed for the first time, aiming to address person retrieval by making full use of combined visual-textual information.

- A scalable automatic triplet synthesis pipeline is presented, which greatly alleviates the difficulties in CPR data annotation. Based on this pipeline, the first million-scale, high-quality and fully synthetic CPR dataset named SynCPR is constructed.

- A new CPR framework, called FAFA is proposed, which significantly improves retrieval performance through fine-grained dynamic alignment and bidirectional masked feature reasoning.

- The first carefully annotated test set named ITCPR is constructed, and extensive experiments validate the effectiveness of our proposed methods.

## 2   Related Work

**Person Retrieval.** Person retrieval primarily comprises two research directions: IPR and TPR. IPR has been extensively explored from various perspectives, including feature extraction [3, 14, 15], metric learning [16], lightweight architecture design [17–19], multi-branch frameworks [1, 20], and attention mechanisms [21, 22]. A related subtask, clothes-changing image person retrieval (CC-IPR) [11], targets identification across outfit variations and has driven the development of specialized datasets [11, 13, 12] and methods [23–25]. In comparison, TPR emerges later but has progressed rapidly. It focuses on aligning visual and textual features within a unified embedding space. Early TPR approaches emphasize global [26–28] and local [29–33] feature extraction and employ cross-modal matching losses [34] but often have difficulty in balancing efficiency and accuracy. More recently, visual-language pretrained (VLP) models [35–38] have significantly improved retrieval performance through carefully designed auxiliary tasks [39, 40, 2] tailored specifically for TPR fine-tuning. However, despite substantial progress, existing approaches still struggle to effectively integrate visual and textual information for precise identification of specific individuals, which remains an essential and practical requirement. To bridge this gap, we propose the CPR task.

**Composed Image Retrieval.** CIR [41–43], as a representative compositional learning task [44, 45], jointly leverages image and textual queries for precise image retrieval. CIR has been extensively applied in fashion [5] and real-world domains [6, 46], fostering diverse image-text fusion and training strategies. However, existing supervised CIR methods [47, 48, 42, 43] heavily depend on annotated triplet datasets, inherently limiting their generalizability. To alleviate reliance on annotation, recent zero-shot CIR (ZSCIR) approaches [49] propose techniques such as image-to-pseudo-text conversion [49, 46, 50, 51] or using LLM-generated target descriptions to reformulate CIR as pure text-to-image retrieval [52, 53]. Compared to CIR, CPR imposes stricter constraints on image relevance and places greater emphasis on fine-grained variations during retrieval. Consequently, existing CIR methods generally struggle to maintain effectiveness under the CPR setting.

**Diffusion Models.** Diffusion models [54, 55] have become the prevailing architecture for image generation, with applications in text-to-image synthesis [56–58], image translation [59–61], and controllable content generation [62–64]. This progress has been accompanied by efficient parameter tuning strategies such as Low-Rank Adaptation (LoRA) [9] and Adapter-based [65] methods, which retain high generation quality while enhancing adaptability. The incorporation of Transformer [66] architectures has led to novel designs like the Diffusion Transformer (DiT) [67], improving scalability and bring about advanced models such as Stable Diffusion 3 [68], PixArt [69], and Flux [8]. Inspired by the above, our work elegantly combines the Flux model with LoRA-based fine-tuning to generate person images that closely resemble visual styles in the real world.

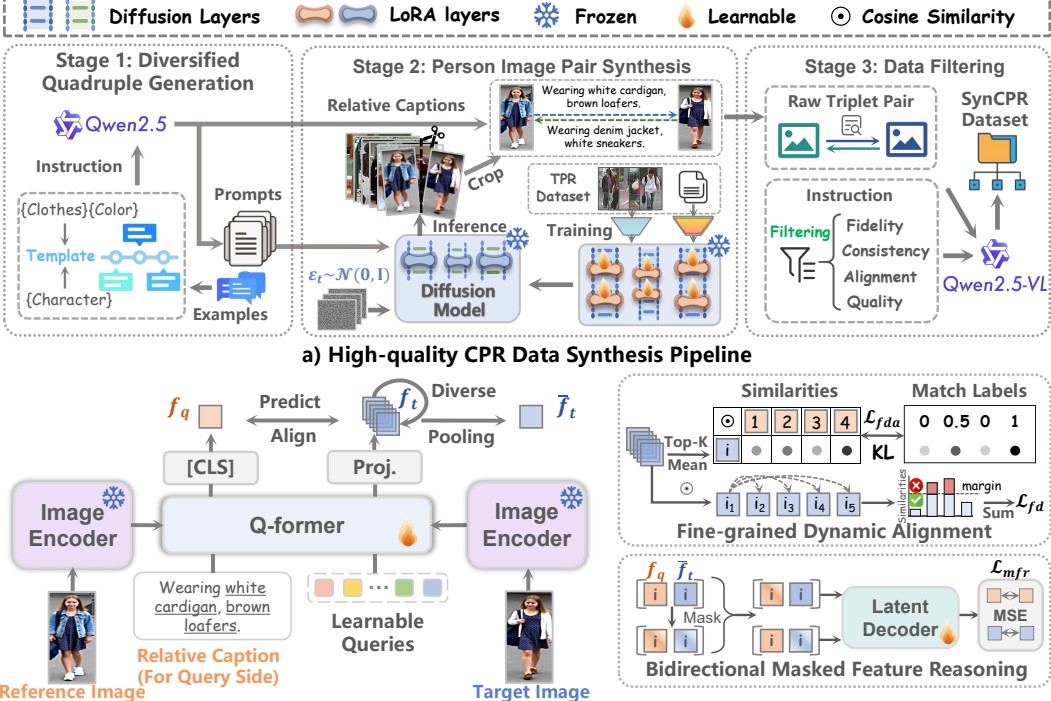

a) High-quality CPR Data Synthesis Pipeline

b) Fine-grained Adaptive Feature Alignment Framework

Figure 2: Overall framework of our method. (a) The pipeline for synthesizing high-quality triplets, consisting of three key stages: generation of text quadruples, synthesis of person image pairs, and data filtering. (b) The structure of FAFA. The left part illustrates the training process of the model, while the right part highlights the key objectives employed by FAFA.

## 3 Method

The overall framework of the proposed CPR method is illustrated in Figure 2, comprising two main components. Section 3.1 introduces the automatic pipeline for synthesizing high-quality CPR data, including textual quadruple generation, identity-consistent image synthesis, and data filtering. Section 3.2 presents the FAFA framework, detailing the model architecture, fine-grained dynamic alignment objectives, bidirectional masked feature reasoning strategies during training, and the inference procedure. To objectively evaluate the proposed method, Section 3.3 outlines the construction of the ITCPR test set.

### 3.1 High-quality CPR Data Synthesis

**Diverse Textual Quadruples Generation.** Considering that there is currently a lack of feasible methods for directly generating multimodal triplet data, we propose decomposing this task into the generation of single-modality triplets first and then expanding them into multimodal form, which effectively alleviates this problem. Specifically, an instruction template $\mathcal{P}(Character, Clothes, Color)$ is designed to guide the LLM [7] (denoted as $\mathcal{G}_{llm}(\cdot)$) to produce textual quadruples. Each quadruple comprises two pairs of textual triplets, as expressed in Equation 1:

$$\mathcal{G}_{llm}(p) \rightarrow \langle T_{I_q}, T_{q \rightarrow t}, T_{t \rightarrow q}, T_{I_t} \rangle, \tag{1}$$

where $T_{I_q}$ and $T_{I_t}$ denote the same person with different outfits or states and will later be used to synthesize images $I_q$ and $I_t$. The relative caption $T_{q \rightarrow t}$ highlights key appearance changes from $I_q$ to $I_t$, while the reverse caption $T_{t \rightarrow q}$ describes changes in the opposite direction, allowing two usable triplets to be constructed from each quadruple. To enhance diversity and avoid repetitive outputs, each instruction $p \sim \mathcal{P}$ includes multiple descriptive elements and randomly selected high-quality annotated examples. Providing these random elements and examples ensures semantic richness, diversity, output quality, and structural stability (see Appendix A.1 for details). A simplified version of the instruction template is shown below:

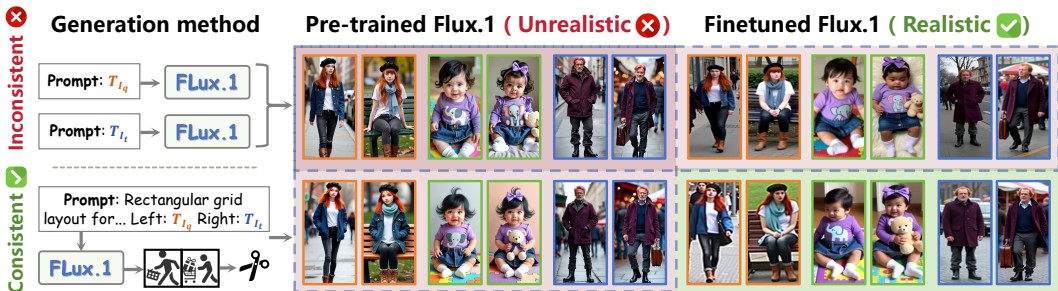

Figure 3: Example pairs of generated person images using different generative models and generation methods under the same text input.

Generate a quadruple satisfying CPR requirements using the following elements: suggested character, clothes, color. Follow the output format and content length from: examples.

**Identity-consistent High-quality Image Synthesis.** As mentioned before, generative models have been widely applied to text-to-image synthesis. However, most of them are oriented towards natural images and portraits, and methods specifically for generating pedestrian images are still rare. Therefore, person images generated by pretrained models often deviate significantly from the style and distribution of real-world person images encountered in retrieval tasks. To address this, we fine-tune the cross-attention layers of DiT [67] using LoRA [9] on the dataset of person image-text pairs:

$$\text{Attention}(Q, K, V) = \text{Softmax}\left(\frac{QK^T}{\sqrt{d}}\right)V, \quad K = \mathbf{W}K\tau\text{txt}(T_{\text{txt}}), \quad V = \mathbf{W}V\tau\text{txt}(T_{\text{txt}}) \quad (2)$$

where $Q$ denotes DiT image features, $\tau_{\text{txt}}(T_{\text{txt}})$ is the text encoder output, and $\mathbf{W}_K$, $\mathbf{W}_V$ are learnable projection matrices. During fine-tuning, only the LoRA components in the cross-attention layers are updated, while all other parameters remain frozen. Given a weight matrix $\mathbf{W} \in \mathbb{R}^{h \times l}$, LoRA introduces trainable matrices $B \in \mathbb{R}^{h \times r}$ and $A \in \mathbb{R}^{r \times l}$, with $r \ll \min(h, l)$, and computes the residual update as $\Delta\mathbf{W} = \beta\gamma BA$, where $\beta$ controls LoRA strength and $\gamma$ is a learnable layer-specific scaling factor. The updated weights are then given by $\mathbf{W}' = \mathbf{W} + \Delta\mathbf{W}$, enabling parameter-efficient adaptation. Training is guided by a flow-matching objective function [70].

Once fine-tuning is complete, as shown in Figure 3, the generative model's inherent consistency capability, that is, the ability to generate coherent elements within a single image, is ingeniously leveraged to synthesize image pairs with consistent identities, which cannot be achieved through independent generation. Specifically, we first define a layout prefix and merge $T_{I_q}$ and $T_{I_t}$ into a unified prompt. Then, this prompt is input to the model to generate a single image with left and right sub-images. The final images $I_q$ and $I_t$ are obtained by cropping:

Rectangular grid layout for left and right images. Each image is independent, ... Left: $T_{I_q}$, Right: $T_{I_t}$.

Furthermore, to maximize textual quadruple utilization, we dynamically adjust $\beta$, generating $n$ image pairs for each textual pair $(T_{I_q}, T_{I_t})$, thus creating $2n$ triplets. Besides, images within the same triplet share a unique ID, while those within groups that share the same relative captions are assigned a common group ID (GID), facilitating label smoothing during training.

**Data Filtering.** To ensure the quality of generated data, the MLLM [10] is employed to evaluate each generated triplet on a scale from 1 to 10 across four criteria: (1) naturalness of individuals in $I_q$ and $I_t$ (excluding resolution and instead focusing on visual realism, noise, and artifact presence); (2) identity consistency between $I_q$ and $I_t$; (3) alignment between images and their corresponding descriptions ($I_q \leftrightarrow T_{I_q}$); and (4) CPR task relevance ($I_q + T_q \rightarrow I_t$). Triplets with an average score below a strict threshold of 8.5 are discarded, leading to the removal of approximately 59% of the data.

Based on this pipeline, a large-scale synthetic dataset named SynCPR is constructed, consisting of 1.15 million high-quality triplets. Further implementation details regarding data synthesis (e.g., complete prompt templates and additional visualization examples) can be found in the Appendix A.

## 3.2 End-to-End Composed Person Retrieval Framework

A new retrieval framework is proposed to achieve end-to-end CPR, where the FAFA is constructed to achieve fine-grained feature alignment.

### 3.2.1 The FAFA Architecture

Inspired by BLIP-2 [38], the proposed FAFA architecture, as shown in Figure 2(b), integrates a frozen image encoder and a lightweight Query Transformer (Q-Former). The Q-Former enables efficient multimodal representation extraction through a trainable query mechanism. It supports two encoding pathways: one is an image-guided path that combines visual and textual inputs, and the other is a purely visual path.

Given an input triplet $\langle I_q, T_q, I_t \rangle$, the frozen image encoder extracts visual features from the reference image $I_q$, which are then combined with the relative caption $T_q$ and fed into the Q-Former. The textual [CLS] token, after passing through a text projection layer, yields the query representation $f_q \in \mathbb{R}^d$. Meanwhile, the target image $I_t$ is processed by the same frozen encoder, and its visual features are routed through the purely visual branch of the Q-Former. The learnable query tokens in this branch are projected along the sequence dimension using a visual projection layer, generating the fine-grained feature representation $f_t = \{f_t(1), f_t(2), \ldots, f_t(N)\} \in \mathbb{R}^{N \times d}$, where $N$ denotes the number of learnable queries and $d$ is the feature dimension.

### 3.2.2 Fine-grained Adaptive Feature Alignment

Fine-grained feature matching is another inherent challenge in the CPR task. To deal with the issue, we propose a fine-grained dynamic alignment mechanism, integrating feature diversity supervision and masked feature reasoning into an end-to-end optimization strategy.

**Fine-grained Dynamic Alignment (FDA).** Unlike conventional contrastive learning methods [35, 71] that focus on global single-feature matching, the proposed approach dynamically aligns multiple fine-grained features from the target image with the query representation. Specifically, for each input triplet, the similarity between the query representation $f_q$ and the set of target fine-grained features $f_t$ is calculated by using dynamic feature selection and aggregation:

$$\text{Sim}(f_q, f_t) = \frac{1}{k} \sum_{i=1}^{k} \text{TopK}_i \left( \left\{ \frac{f_q^\top f_t(j)}{\|f_q\| \cdot \|f_t(j)\|} \right\}_{j=1}^{N} \right) \tag{3}$$

where $\text{TopK}_i(\cdot)$ denotes the $i^{th}$ highest similarity score. This mechanism allows the model to adaptively select the most relevant fine-grained features for improved precision. During training, distribution matching and label smoothing are incorporated to enhance contextual alignment. For batch size $B$, the ground-truth matching probability is defined as: $q_{i,j} = \frac{y_{i,j}}{\sum_{k=1}^{B} y_{i,k}}$, where $y_{i,j} = 1$ for exact matches (with the same ID), $y_{i,j} = \alpha, \alpha \in (0, 1)$ for partial matches (with the same GID), and $y_{i,j} = 0$ for unmatched pairs. The predicted distribution is normalized via softmax: $p_{i,j} = \frac{\exp(\text{Sim}(f_q^i, f_t^j)/\tau)}{\sum_{k=1}^{B} \exp(\text{Sim}(f_q^i, f_t^k)/\tau)}$, where $\tau$ is a temperature parameter. Then, the query-to-target alignment loss is defined as:

$$\mathcal{L}_{q2t} = \frac{1}{B} \sum_{i=1}^{B} \text{KL}(\mathbf{p_i} | \mathbf{q_i}) = \frac{1}{B} \sum_{i=1}^{B} \sum_{j=1}^{B} p_{i,j} \log \left( \frac{p_{i,j}}{q_{i,j} + \epsilon} \right) \tag{4}$$

where $\text{KL}(\cdot | \cdot)$ represents Kullback–Leibler divergence and $\epsilon$ ensures numerical stability. The reverse loss $\mathcal{L}_{t2q}$ is computed analogously by interchanging $f_q$ and $f_t$, leading to the overall alignment loss: $\mathcal{L}_{fda} = \mathcal{L}_{q2t} + \mathcal{L}_{t2q}$.

**Feature Diversity (FD) Supervision.** To reduce redundancy, a feature dispersion loss is introduced:

$$\mathcal{L}_{fd} = \frac{1}{N(N-1)} \sum_{i \neq j} \max \left( \frac{f_t(i)^\top f_t(j)}{|f_t(i)| \cdot |f_t(j)|} - m, 0 \right) \tag{5}$$

where $m$ sets the maximum cosine similarity, encouraging diversity among internal representations.

**Masked Feature Reasoning (MFR).** To exploit complementary information between reference images and relative text, a bidirectional MFR strategy is proposed. Specifically, the random masking operation (30%) is applied to $f_q$ and the average pooled target image feature $\bar{f}_t$, thus producing

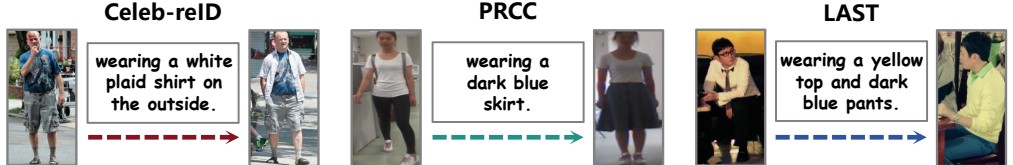

Figure 4: Some representative examples from the ITCPR dataset.

masked features $\tilde{f}_q$ and $\tilde{f}_t$. The four features are jointly fed into a lightweight decoder $\Phi$ to minimize reconstruction loss:

$$\mathcal{L}_{\mathrm{mfr}} = \mathbb{E}_{(f_q, \bar{f}_t) \sim \mathcal{B}} \left[ |f_q - \Phi([\bar{f}_t, \tilde{f}_q])|_2^2 + |\bar{f}_t - \Phi([f_q, \tilde{f}_t])|_2^2 \right] \tag{6}$$

This loss drives the model to recover complete representations and enhance cross-modal alignment. Finally, by combining the above three components, the overall training objective is formulated as: $\mathcal{L} = \mathcal{L}_{fda} + \lambda_1 \mathcal{L}_{fd} + \lambda_2 \mathcal{L}_{mfr}$, where $\lambda_1$ and $\lambda_2$ are balancing weights for the auxiliary loss terms.

### 3.2.3 Inference Workflow

During inference, the fine-grained feature sets for all target images in the retrieval dataset are pre-extracted and stored as $\mathcal{V} = f_{t\ i=1}^{i\ N_t}$. Given a combined query feature $f_q$, the similarity between it and each $f_t^i$ is computed using the same dynamic alignment method employed during training, thus ensuring efficient and reliable retrieval of the most relevant target images.

### 3.3 ITCPR Dataset

To objectively evaluate CPR methods, we manually construct the ITCPR dataset. Each triplet contains a reference image and a target image sharing the same identity, selected from public clothes-changing datasets including Celeb-reID [11], PRCC [13], and LAST [12], ensuring identity consistency despite variations in clothing or background. Each triplet also includes a relative caption explicitly highlighting differences between the two images, requiring models to jointly leverage visual and textual information for accurate retrieval. To ensure evaluation reliability, gallery images are carefully reviewed to eliminate potential false-negative cases. Ultimately, ITCPR contains 2,225 annotated triplets, comprising 2,202 unique query combinations $(I_q, T_q)$ from 1,199 identities. The gallery consists of 20,510 person images, among which 2,225 correspond directly to queries. Representative examples are illustrated in Figure 4.

## 4 Experiments

### 4.1 Experimental Setup

**Datasets.** For data generation, we fine-tune Flux.1 [8] on the training split of CUHK-PEDES [4], a widely-used real-world person dataset containing 68,126 manually annotated image-text pairs. For the CPR task, FAFA is trained on 1.15 million filtered high-quality triplets from SynCPR, and evaluations are conducted on the manually annotated ITCPR dataset. Detailed descriptions of all datasets are provided in the Appendix B.

**Choice of Dataset for Fine-tuning.** For fine-tuning the model, we chose CUHK-PEDES over other datasets such as UFine6926 [72], ICFG-PEDES [73], and RSTPReid [74]. CUHK-PEDES was selected primarily because of its greater scene diversity, encompassing images from five surveillance datasets that represent a wide range of real-world scenarios, including urban environments and public spaces. This diversity is crucial for ensuring that the model generalizes well across various contexts, which is essential for practical person retrieval tasks. In contrast, ICFG-PEDES and RSTPReid, which mainly focus on constrained environments like parking lots (MSMT17 [75]), lack the same level of visual variation, potentially limiting the model's adaptability to real-world scenarios. Furthermore, while UFine6926 offers more fine-grained text-image pairings, its higher image quality and controlled video sources do not provide the same environmental diversity as CUHK-PEDES, which could lead to overfitting. By fine-tuning on CUHK-PEDES, we strike a balance between rich textual descriptions

Table 1: **Comparison of methods across different domains and settings.** For all domains other than CPR, models are trained on the most representative dataset within each domain.

| Domain | Method | Ref. | Pretraining Data | Setting | Rank-1 | Rank-5 | Rank-10 | mAP |
|---|---|---|---|---|---|---|---|---|
| IPR | TransReID [76] | ICCV21 | Market-1501 [77] | *Image-only* | 7.27 | 17.30 | 22.75 | 12.57 |
| | SOLIDER [78] | CVPR23 | | | 8.45 | 18.48 | 23.89 | 13.74 |
| | CLIP-ReID [79] | AAAI23 | | | 7.95 | 18.12 | 22.75 | 13.31 |
| CC-IPR | CAL [80] | CVPR22 | LTCC [81] | *Image-only* | 9.86 | 22.34 | 29.20 | 16.45 |
| | FIRe2 [82] | TIFS24 | | | 10.76 | 22.84 | 29.29 | 17.00 |
| TPR | RaSa [83] | IJCAI23 | CUHK-PEDES [4] | *Text-only* | 28.02 | 49.23 | 57.77 | 38.04 |
| | IRRA [2] | CVPR23 | | | 26.39 | 46.46 | 56.27 | 36.13 |
| | RDE [84] | CVPR24 | CUHK-PEDES [4] | *Image-only* | 6.31 | 13.78 | 18.46 | 10.43 |
| | | | | *Text-only* | 26.43 | 47.41 | 56.45 | 36.35 |
| | | | | *Image + Text* | 29.79 | 51.82 | 60.49 | 40.10 |
| Fuse | SOLIDER + RaSa | - | - | *Image + Text* | 30.97 | 52.86 | 61.81 | 41.22 |
| | FIRe2 + RaSa | - | | | 32.89 | 54.27 | 62.03 | 42.16 |
| ZSCIR | Pic2Word [49] | CVPR23 | CC3M [85] | *Combination* | 21.21 | 37.15 | 44.51 | 29.11 |
| | CoVR-BLIP [86] | AAAI24 | WebVid-CoVR [86] | | 26.75 | 47.68 | 56.36 | 36.49 |
| | LinCIR (ViT-G) [87] | CVPR24 | - | | 23.93 | 44.46 | 53.18 | 33.95 |
| CIR | CaLa [47] | SIGIR24 | CIRR [6] | *Combination* | 24.02 | 44.64 | 53.45 | 34.08 |
| | | | **SynCPR (Ours)** | | 39.33 | 60.85 | 68.66 | 49.29 |
| | SPRC [48] | ICLR24 | CIRR [6] | *Combination* | 25.07 | 45.73 | 54.50 | 35.05 |
| | | | **SynCPR (Ours)** | | 42.27 | 61.81 | 69.35 | 51.62 |
| CPR | **FAFA (Ours)** | - | **SynCPR (Ours)** | *Combination* | **46.54** | **66.21** | **73.12** | **55.60** |

*\*__Bold__ indicates the best performance; Underline indicates the second best.*

and diverse visual settings, ensuring the model's robustness when confronted with the variety of challenges encountered in real-world person retrieval tasks.

**Implementation Details.** All experiments are conducted using two H800 GPUs. During the SynCPR construction process, we adopt Qwen2.5-70B [7] as the LLM to generate textual quadruples, and use Flux.1 [8] as the base image generation model. This model is fine-tuned by LoRA [9] with its rank $r = 64$, and we set $\beta = 1$ to generate five identity-consistent image pairs per quadruple in the most realistic style. Another five image pairs are generated using random values of $\beta \in (0, 1)$ to ensure stylistic diversity. Qwen2.5VL-32B [10] is employed for data filtering. For training the FAFA framework, we set the total number of epochs to 10 and use a batch size of 256. The soft label strength in FDA is set to $\alpha = 0.5$, the number of selected fine-grained features is $k = 6$, and $\tau = 0.02$. The margin parameter $m$ in $\mathcal{L}_{fd}$ is set to 0.5. The loss balancing hyperparameters are set to $\lambda_1 = 1$ and $\lambda_2 = 0.5$. All comparison methods are implemented using the optimal settings reported by them. Additional implementation details can be found in the Appendix C.1.

**Evaluation Metrics.** Retrieval performance is measured using Rank-k accuracy and mean average precision (mAP). Rank-k indicates the probability of correct matches in top-k retrievals, while mAP averages precision across all queries.

## 4.2 Results

To objectively evaluate FAFA and the SynCPR dataset, we extensively compare recent approaches from person retrieval and composed image retrieval. The compared methods are categorized into four settings based on input types: 1) *Image-only*, which relies solely on the reference image and retrieves targets via the visual encoder; 2) *Text-only*, which uses only relative captions and retrieves targets through cross-modal alignment; 3) *Image + Text*, which calculates similarity scores separately via the first two methods and then retrieves targets using their average; and 4) *Combination*, which simultaneously inputs both reference image and relative caption into the model for target retrieval. As shown in Table 1, our method consistently outperforms others across all settings. Specifically, directly applying IPR methods yields the lowest performance due to clothing variations between reference and target images. Even CC-IPR methods trained explicitly on clothes-changing datasets struggle due to limited generalization. In contrast, TPR methods achieve relatively better results, as

Table 2: **Ablation experiments on each component of FAFA.** To validate the effectiveness of FDA, we additionally introduce the image–text contrastive loss (ITC) [71] for comparison.

| No. | Components | | | | | ITCPR Dataset | | | |
| --- | --- | --- | --- | --- | --- | --- | --- | --- | --- |
| | SynCPR | ITC | FDA | FD | MFR | Rank-1 | Rank-5 | Rank-10 | mAP |
| 1 | ✓ | ✓ | | | | 41.33 | 61.72 | 68.94 | 50.94 |
| 2 | ✓ | | ✓ | | | 45.04 | 64.90 | 72.21 | 54.41 |
| 3 | ✓ | | ✓ | ✓ | | 46.05 | 65.85 | 73.02 | 55.49 |
| 4 | ✓ | | ✓ | | ✓ | 45.78 | 65.58 | 72.62 | 55.13 |
| 5 | ✓ | | ✓ | ✓ | ✓ | **46.54** | **66.21** | **73.12** | **55.60** |

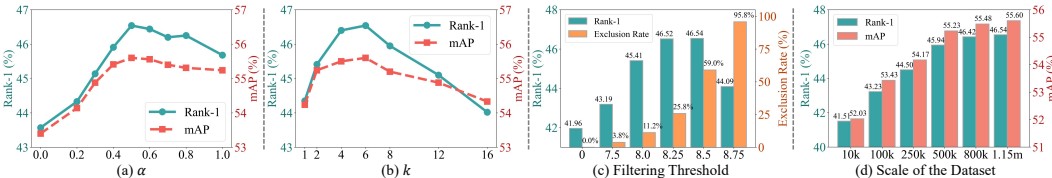

Figure 5: Sensitivity analysis of FAFA on hyperparameters and analysis of the SynCPR dataset.

the relative captions inherently match target images, although some visual information is missing. Among baseline approaches excluding our method, the *Image + Text* strategy achieves the best results, validating the rationality of our ITCPR dataset and emphasizing the necessity of combining visual and textual queries for optimal retrieval.

For CIR methods, although inherently designed for joint image-text queries, their training generally targets natural images involving significant visual modifications, thus lacking fine-grained retrieval capability required by CPR tasks. Notably, supervised CIR methods trained on original CIR datasets perform worse than certain ZSCIR methods on our task, underscoring the need for CPR-specific datasets and methods. Training supervised CIR methods on our SynCPR dataset significantly improves retrieval performance to a practical level. Furthermore, integrating our fine-grained retrieval framework FAFA with SynCPR further substantially enhances retrieval accuracy, confirming the indispensable roles of both the proposed dataset and FAFA.

## 4.3 Ablation Study

In this section, we conduct comprehensive ablation experiments to investigate the contribution of each component within the FAFA framework. Additionally, we discuss the impact of key hyperparameters in both the FAFA model and the data generation process.

**FAFA Model.** We train variants of the FAFA model with different components on the SynCPR dataset and evaluate their performance on the ITCPR test set. As shown in Table 2, experimental results demonstrate that due to the specific nature of the CPR task, employing our proposed fine-grained dynamic alignment strategy can substantially improve retrieval performance. Moreover, both supervision strategies, namely the FD strategy for enhancing feature diversity and the MFR strategy for capturing complementary features, contribute effectively to performance gains. The FAFA model equipped with all components achieves the best overall performance.

**Hyperparameters of FAFA.** Figures 5(a) and 5(b) illustrate the impact of two critical hyperparameters in our proposed FAFA model, namely the soft label strength $\alpha$ and the number of selected fine-grained features $k$ in FDA, on the retrieval performance. Regarding $\alpha$, lower values mean that the triplets generated from the same textual data will be treated more negatively, thus have an adverse impact on FAFA's semantic understanding. Conversely, higher values will weaken FAFA's ability to maintain identity consistency. This observation is consistent with our experimental results: as $\alpha$ increases, the retrieval performance initially improves and subsequently declines, achieving optimal performance when $\alpha = 0.5$. Similarly, the number of fine-grained features $k$ also exhibits a comparable trend, and the optimal performance can be obtained when $k = 6$. This also aligns with expectations, because smaller $k$ values will restrict the involvement of sufficient fine-grained features in retrieval, whereas excessively large $k$ values may make the training process too homogenized, thus are not suitable for retrieval tasks that require distinctive feature representations.

**SynCPR Dataset.** Figure 5(c) presents the influence of applying various scoring thresholds on retrieval performance and data filtration ratio after generating all triplet data. Without any filtering, the potential noise in the dataset negatively impacts the FAFA training process, and consequently reduces retrieval performance. The optimal retrieval performance is observed when the threshold is set at 8.25 and 8.5. To enhance training efficiency and ensure the high quality of the SynCPR dataset, we finally adopt the latter. Furthermore, we perform sampling on the retained 1.15 million high-quality triplets via GID to validate the appropriate scale of the SynCPR dataset. As shown in Figure 5(d), as the dataset size increases, the retrieval performance improves rapidly. When the number of samples exceeds 500k, the marginal gains gradually diminish, and it saturates when the number of samples reaches approximately 800k. This confirms that our SynCPR dataset containing 1.15 million triplets is large and challenging enough to train better CPR models and is also convenient for comparison with our baseline method.

## 5 Conclusion

We introduce a practically significant task of composed person retrieval. Firstly, we put forward a scalable synthetic pipeline to address the data scarcity problem, and construct a high-quality SynCPR dataset at million scale. Secondly, a novel FAFA framework is introduced to enhance fine-grained retrieval accuracy. Extensive experiments on the newly annotated ITCPR benchmark confirm the significant superiority of our approach over the existing IPR, TPR, and CIR methods. Future work will explore composed person retrieval based on multiple images and multiple textual descriptions, as well as retrieval under open-set conditions.

**Ethical Considerations.** While the CPR task holds significant promise for applications such as locating missing individuals, it also raises critical ethical concerns, particularly regarding privacy and the potential for surveillance misuse. The ability to track individuals across different locations introduces privacy risks, which can be mitigated through various safeguards. For instance, invisible digital watermarks have been embedded in the images of the generated SynCPR dataset to ensure traceability, and access is restricted to academic use under responsible-use agreements. Additionally, biases inherent in synthetic data generated by LLMs have been addressed by ensuring substantial diversity in the generated data. Statistical information and visual representations in Appendix B.2 of the appendix effectively demonstrate the demographic diversity of the SynCPR dataset, encompassing various genders, ages, and ethnicities. These measures ensure the responsible use of the proposed method, adhering to the principle of "Tech for Good" while addressing potential societal risks.

**Limitations**

While the proposed CPR task demonstrates significant potential, several limitations remain. The SynCPR dataset, although highly diverse, relies on synthetic data, which may still introduce a domain gap when applied to real-world scenarios. Despite efforts to minimize this gap through adjustments in the generation strategy and fine-tuning of the generative model, the synthetic nature of the training data may not fully capture all the variations found in real-world images. Additionally, the ITCPR test set primarily focuses on clothing changes, which limits the model's ability to generalize to other types of variations, such as changes in scenes or hairstyles. Expanding the dataset to encompass a broader range of person-related variations will be an important area for future improvement. Furthermore, the current FAFA framework and the ITCPR test set focus mainly on scenarios where only a single image and textual description are provided, which may not align with more complex real-world situations. Addressing this limitation will be a key focus for future work.

## Acknowledgements

This work was supported by the Beijing Advanced Innovation Center for Future Blockchain and Privacy Computing (GJJ-24-021) and the BUPT Innovation and Entrepreneurship Support Program (2025-YC-T043).

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

# Appendix

## A More Details for High-Quality Triplet Synthesis

The triplet data required for Composed Person Retrieval (CPR) consists of three key elements: a reference image $I_q$, a relative caption $T_q$, and a target image $I_t$. Two significant challenges hinder the complete synthetic generation of such triplets. Firstly, generating a pair of person images $(I_q, I_t)$ consistent with real-world distributions while preserving identity. Secondly, providing accurate textual descriptions of relative changes between the two images. To address these challenges, as shown in Figure S6, we effectively divide the data synthesis process into three steps. First, a Large Language Model (LLM) [7] generates textual data comprising descriptions for synthesizing image pairs and relative captions. Second, generative models [8] utilize the textual descriptions to produce realistic and identity-consistent image pairs, thereby forming the required triplets. To ensure realism in the generated images, we further fine-tune the generative models. Finally, a multimodal large language model (MLLM) [10] evaluates the synthesized triplets across multiple dimensions, filtering out lower-quality data.

### A.1 Diverse Textual Quadruples Generation

In this step, we simplify the multimodal triplet generation objective $(I_q, T_q, I_t)$ to purely textual quadruple generation $(T_{I_q}, T_{q \to t}, T_{t \to q}, T_{I_t})$ using an LLM. Each quadruple comprises a reference description $T_{I_q}$, a target description $T_{I_t}$, a relative caption describing changes from $T_{I_q}$ to $T_{I_t}$ ($T_{q \to t}$), and another describing the reverse changes ($T_{t \to q}$). To achieve this, we select QWen2.5-72B [7] as the LLM and carefully design structured instructions to generate quadruples meeting the desired criteria.

The instruction format, illustrated in Figure S6, initially provides an overview of the task and fundamental requirements for the LLM. It then specifies detailed guidelines for generating each element within the quadruple. Additionally, the instructions include three high-quality example outputs randomly selected from 100 manually annotated cases to enhance the quality and stability of the LLM outputs. Random sampling of examples promotes diversity in instructions, preventing repetitive outputs caused by similar inputs.

Notably, the instructions suggest the primary character, clothing items, and colors, randomly drawn from candidate lists. These lists are derived from relevant datasets [4, 73, 74] containing person descriptions, supplemented by additional related elements. Such a design ensures generated data closely aligns with real-world distributions while maximizing its diversity and comprehensiveness. The LLM subsequently generates structured prompts for image synthesis and composed retrieval based on these provided elements.

### A.2 Identity-consistent High-quality Image Synthesis

Once the textual quadruples $(T_{I_q}, T_{q \to t}, T_{t \to q}, T_{I_t})$ are obtained, we convert $T_{I_q}$ and $T_{I_t}$ into their corresponding images, $I_q$ and $I_t$, thus forming the desired triplet data. For this purpose, we adopt FLUX.1 [8], an advanced generative model, as the base model and fine-tune it using Low-Rank Adaptation (LoRA) [9] on a text-based person retrieval (TPR) dataset [4] to generate person images consistent with real-world distributions. Due to the inherent randomness in diffusion model image generation, a critical challenge remains ensuring identity consistency between paired images. However, diffusion models intrinsically possess the capability to generate two identical or similar objects within one image. We leverage this internal consistency capability by generating two sub-images within a single image, thereby ensuring detailed consistency of shared elements in $I_q$ and $I_t$. To achieve this, we specifically design the image generation prompt template to include two equally sized sub-images with the same identity.

During this stage, prompts generated by the LLM are input into FLUX.1 to produce images with a resolution of $400 \times 400$. This configuration allows each generated image to be split into two sub-images ($192 \times 384$), forming the pair $(I_q, I_t)$. This padding strategy mitigates inaccuracies that may arise during image generation and avoids artifacts caused by image cropping. As depicted in Figure S6, this method effectively maintains identity consistency between the sub-images while varying their appearances and states, demonstrating clear advantages over separate generation. Consequently,

**Character List**

*man, woman, boy, girl, teenager, elderly man, elderly woman, child, baby, toddler, young adult, middle-aged man, middle-aged woman, student, teacher, doctor, nurse, chef, engineer, office worker, police officer, firefighter, farmer, artist, musician, athlete, construction worker, salesperson, scientist, pilot, driver, barista, tourist, shopper, cyclist, jogger, hiker, swimmer, dancer, yoga practitioner, gardener, photographer, traveler, waitress, street vendor, businessman, student with backpack, person in wheelchair, siblings, bride, groom, bride with wedding dress, groom in suit, person with pet dog, person holding umbrella, person wearing headphones, military officer, paramedic.*

**Color List**

*red, green, blue, yellow, cyan, magenta, black, white, gray, brown, orange, purple, pink, beige, ivory, navy, teal, maroon, olive, lime, gold, silver, bronze, amber, peach, turquoise, lavender, coral, indigo, plum, salmon, mint, khaki, chocolate, crimson, violet, emerald, jade, aquamarine, rose, charcoal, cream, tan, burgundy, scarlet, chartreuse, cobalt, periwinkle, ruby, sapphire, amethyst, topaz, fuchsia, blush, canary, copper, denim, orchid, pearl, rust, sage, seafoam, sepia, slate, tangerine, ultramarine, vermillion, wine, forest green, sky blue, ocean blue, sand, desert, sunset orange, midnight blue, stone gray, grass green, cloud white, earth brown, seaweed green, lavender field, mountain gray, rose gold, platinum, steel, onyx, diamond, ruby red, emerald green, sapphire blue, amethyst purple, opal, topaz yellow, ash, pewter, slate gray, graphite, smoke, dove gray, stone, taupe, off-white, eggshell, neon green, neon pink, neon yellow, neon orange, electric blue, hot pink, pastel blue, pastel pink, pastel yellow, pastel green, baby blue, baby pink, fluorescent green, fluorescent orange, royal blue, mustard, apricot, cerulean, persimmon, mauve, ochre, ebony, ebony black, jade green, carnation pink, raspberry, peacock blue, mandarin, brick red, bubblegum pink.*

**Clothes List**

*T-shirt, shirt, sweater, hoodie, tank top, blouse, polo shirt, long-sleeve shirt, cardigan, crop top, sweatshirt, vest, jeans, shorts, skirt, trousers, leggings, sweatpants, cargo pants, chinos, denim skirt, mini skirt, pleated skirt, cargo shorts, jacket, coat, blazer, windbreaker, parka, trench coat, leather jacket, denim jacket, bomber jacket, puffer jacket, raincoat, sneakers, sandals, boots, loafers, high heels, flats, oxford shoes, running shoes, hiking boots, slip-on shoes, espadrilles, ankle boots, glasses, sunglasses, scarf, hat, baseball cap, beanie, backpack, handbag, belt, watch, necklace, earrings, bracelet, gloves, hairpin, tie, bow tie, umbrella, headphones, fanny pack, dress, suit, tuxedo, evening gown, sportswear, yoga pants, swimsuit, bikini, apron, uniform, lab coat, chef's hat, striped pattern, floral print, plaid shirt, polka dots, plain color, graphic print, checkered design, rolled-up sleeves, belted waist.*

↓ **Random**     ↓ **Random**     ↓ **Random**

*Please help me provide a prompt with the same structure as my example but different content. Structurally, first provide two detailed descriptions of the appearance of a single person's image, labeled as Prompt1 and Prompt2. Ensure both describe the same person but with partially different outfits (backgrounds/actions may vary, and note that backgrounds/actions are relatively less important compared to appearance, so you can choose to omit relevant parts when describing). When Prompt1 is given, Relative Prompt1 should be a relative description or abbreviated version of Prompt2, only mentioning outfit differences from Prompt1 (omit identical elements). Similarly, Relative Prompt2 should describe only the differences between Prompt1 and Prompt2. Three examples: **{Examples}** Note: The output must describe a '**{Character}**'. At least one prompt must mention '**{Clothes}**', and include at least one '**{Color}**' clothing/accessory. The final output must be logical, strictly follow the example structure, avoid duplicating any example's content, meet all requirements above, and ensure sentence lengths are similar to the examples.*

**LLM Instructions**

**Examples**

🗨 Qwen2.5

**Step 1: Diversified Quadruple Generation**

**LLM Output**

**Prompt1 ($T_{I_q}$):** *A woman with black hair is wearing a teal tank top, black shorts, and white sandals. She is carrying a black crossbody bag.*
**Prompt2 ($T_{I_t}$):** *A woman with black hair is wearing a teal tank top, black shorts, and black ankle boots. She is carrying a white tote bag.*
**Relative Prompt1 ($T_{q→t}$):** *Wearing black ankle boots, carrying a white tote bag.*
**Relative Prompt2 ($T_{t→q}$):** *Wearing white sandals, carrying a black crossbody bag.*

*Rectangular grid layout for left and right images. Each image is independent, shown in photorealistic style. Left: $T_{I_q}$ Right: $T_{t→q}$*

**Generation Prompt**

**Step 2: Constant Image Pair Synthesis**

Dataset    Training    Inference

**CUHK-PEDES**

Diffusion Model

Wearing black ankle boots, carrying a white tote bag.

Wearing white sandals, carrying a black crossbody bag.

$I_q$     $T$     $I_t$

**Step 3: Data Filtering**

**MLLM Instructions**

Qwen2.5-VL

**Scores**

*'Quality': 9,*
*'Consistency': 8,*
*'Align': 10,*
*'Relative_prompt_quality': 9*

→ ✅ SynCPR

*Evaluate two synthetic images (possibly surveillance-style; ignore image sharpness) depicting the same person with different outfits based on four criteria (1-10 scale). Background: Both images are generated with corresponding prompts and mutual modification descriptions. Evaluate these two images across four dimensions (1-10 scale, 1=worst, 10=best). First dimension **'Quality'**: Assess if both images depict plausible human figures without border artifacts or incoherent compositions, ignoring sharpness. Higher scores indicate natural, well-composed figures. Second dimension **'Consistency'**: Evaluate if $I_q$ and $I_t$ represent the same person despite differing outfits/poses/scenes. Higher scores mean stronger facial/body feature consistency. Third dimension **'Align'**: Check how accurately $I_q$ matches $T_{I_q}$ and $I_t$ matches $T_{I_t}$, verifying objects, colors, and scene elements. Higher scores reflect precise text-image alignment. Fourth dimension **'Relative_prompt_quality'**: Judge if $T_{q→t}$ accurately modifies $I_q$ into $I_t$ and whether $T_{t→q}$ accurately reverses this transformation from $I_t$ to $I_q$. Deduct points for incorrect/unnecessary changes. Higher scores mean prompts precisely capture mutual differences while preserving shared elements. Output format: **'Quality': score, 'Consistency': score, 'Align': score, 'Relative_prompt_quality': score**. Use single quotes. Only provide the scores dictionary, no explanations.*

Figure S6: Pipeline of high-quality CPR triplet construction with detailed instruction design.

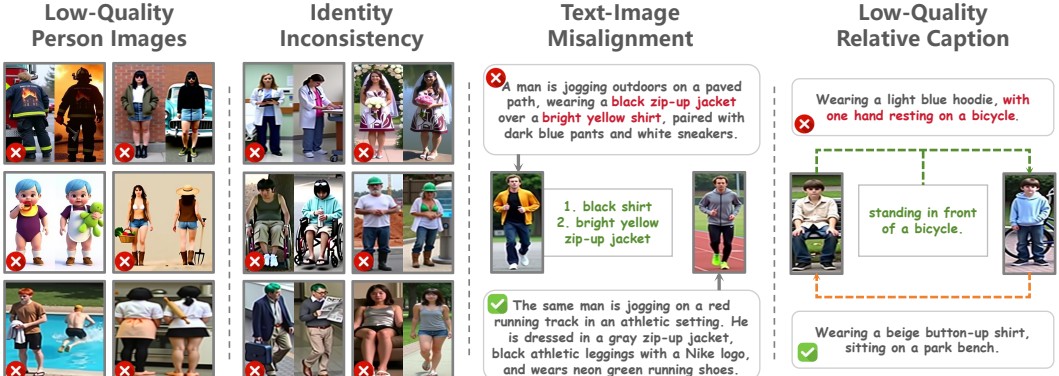

Figure S7: Representative examples of samples filtered out during the data filtering process. From left to right, each panel corresponds to one of the four evaluation dimensions, and the samples are excluded due to low scores in their respective dimensions.

we acquire the desired dataset, and swapping the reference and target images yields two sets of triplet annotations. By dynamically adjusting the LoRA strength $\beta$, each textual quadruple generates ten image pairs. Images with identical relative captions are defined under the same group identity (GID), thus forming strong positive samples within triplets and weak positive samples within groups, with other instances treated as negative samples.

## A.3 Data Filtering

To further ensure the quality of the synthesized data, we employ Qwen2.5-VL 32B [10] combined with carefully designed instructions to score and filter all generated data. Four dimensions are considered: first, assessing the fidelity of person images $I_q$ and $I_t$, focusing on naturalness, noise, and artifacts while ignoring image clarity; second, evaluating identity consistency between $I_q$ and $I_t$; third, assessing the alignment between images and their descriptions (e.g., $I_q \leftrightarrow T_{I_q}$); fourth, evaluating the overall quality of the triplet ($I_q + T_q \rightarrow I_t$), where higher scores indicate the ability to accurately infer $I_t$ from the combination of $I_q$ and $T_q$, with minimal overlap and high complementarity. Qwen2.5-VL rates each dimension from 1 to 10, and the final score is the average of these dimensions. Triplets scoring higher than 8.5 are retained and included in the SynCPR dataset. Representative examples of discarded low-quality data are shown in Figure S7.

# B Additional Datasets Details

## B.1 ITCPR Dataset

In contrast to existing CIR datasets [5, 6], where reference and target images only need to be loosely related, the CPR datasets subject to the constraint that both of them depict the same person. Therefore, when constructing the ITCPR dataset, we ask the selected images to have the same identity, but wear different clothes or be in different scenes. In our implementation, publicly available clothes-changing datasets such as Celeb-reid [11], PRCC [13], and LAST [12] are utilized as our image sources.

### B.1.1 Dataset Annotation Process

The annotation process, as shown in Figure S8, primarily consists of three steps. The first step involves selecting identities from the image data sources that have multiple images with different outfits, ensuring a diverse selection for subsequent steps. In the second step, a pair of images associated with each chosen identity is selected and denoted as $I_q$ and $I_t$. It is worth noting that, ideally, these two images should depict partially matching outfits, allowing $I_q$ to provide additional clothing-related information beyond facial features and body posture. This additional clothing information is not mentioned in the corresponding annotation $T_q$, ensuring that CPR methods can only correctly identify $I_t$ by utilizing both $I_q$ and $T_q$. Once the image pair is selected, the process moves to the third step, where manual annotations are created to specifically capture the differences between $I_q$ and $I_t$. For

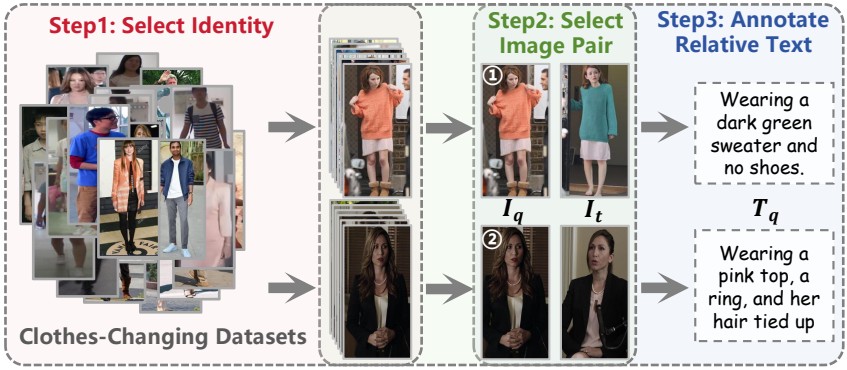

Figure S8: The annotation process of the ITCPR dataset. The annotation process can be summarized in three steps: the first step is selecting identities from the clothes-changing datasets, the second step is choosing pairs of reference and target images for each identity, and the third step is manually annotating the relative captions.

instance, as shown in case 1 of Figure S8, if the skirt is the same in both the target image and the reference image, it does not need to be described; the annotation focuses only on differences in the top and shoes. After manually annotating $T_q$, a complete triplet annotation process is finalized. Repeating this process, a batch of triplets $(I_q, T_q, I_t)$ can be generated for testing CPR methods.

### B.1.2  Re-Annotation of the ITCPR Dataset

The gallery contains a large number of noisy images, which may introduce false negatives. For example, for certain queries, some images may be potential ground truth but remain unlabeled. Including such cases would reduce the reliability of the evaluation metrics. To address this issue, all images in the gallery are screened, and any false negative images identified in the dataset are re-annotated. The re-annotation process is illustrated in Figure S9. After completing the dataset annotation and adding noise images to the gallery, we use a well-trained visual encoder [78] to search for the most similar images for each target image, followed by manual inspection and verification. Through this approach, we effectively eliminate false negatives in the ITCPR dataset.

### B.1.3  Statistics of the ITCPR Dataset

In summary, ITCPR comprises a total of 2,225 annotated triplets. These triplets encompass 2,202 unique combinations $(I_q, T_q)$ as queries. ITCPR contains 1,151 images and 512 identities from Celeb-reID [11], 146 images and 146 identities from PRCC [13], and 905 images and 541 identities from LAST [12]. In the target gallery, there are a total of 20,510 images of persons from the three datasets, with 2,225 corresponding ground truths for the queries. The textual annotations have an average sentence length of 9.54 words. The longest sentence contains 32 words, while the shortest sentence only contains 3 words. These annotations are exclusively designated for testing in the ZS-CPR task, which expects to achieve substantial performance without utilizing any data from the three datasets mentioned above.

### B.2  SynCPR Dataset

Using our proposed automated construction pipeline, we successfully build the SynCPR dataset, which is a fully synthetic dataset specifically designed for the composed person retrieval task. In the first stage, we utilize Qwen2.5-70B [7] to generate a total of 140,500 textual quadruples. In the second stage, by employing fine-tuned LoRA [9] combined with Flux.1 [8] and setting $\beta = 1$ for the most realistic person image style, we generate five image pairs per quadruple. Additionally, we create another five image pairs using randomly selected $\beta \in (0, 1)$, ensuring diverse styles across generated images. Combining these images with two relative captions from each quadruple yields a total of 2,810,000 valid triplets. In the third stage, under stringent data filtering criteria, 1,153,220 high-quality triplets are retained. Among the retained samples, a total of 177,530 unique GIDs are involved. The average length of the relative caption sentences is 13.3 words, excluding punctuation.

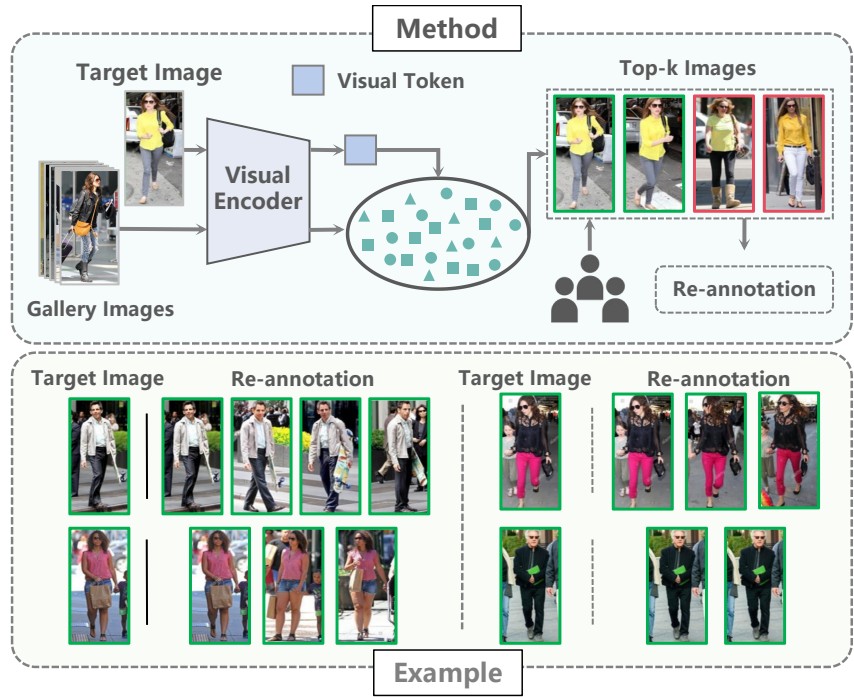

Figure S9: False negative elimination scheme in ITCPR. **Top**: The method of eliminating false negative images and adding annotations in the dataset. **Bottom**: Examples of false negative images re-annotated in ITCPR.

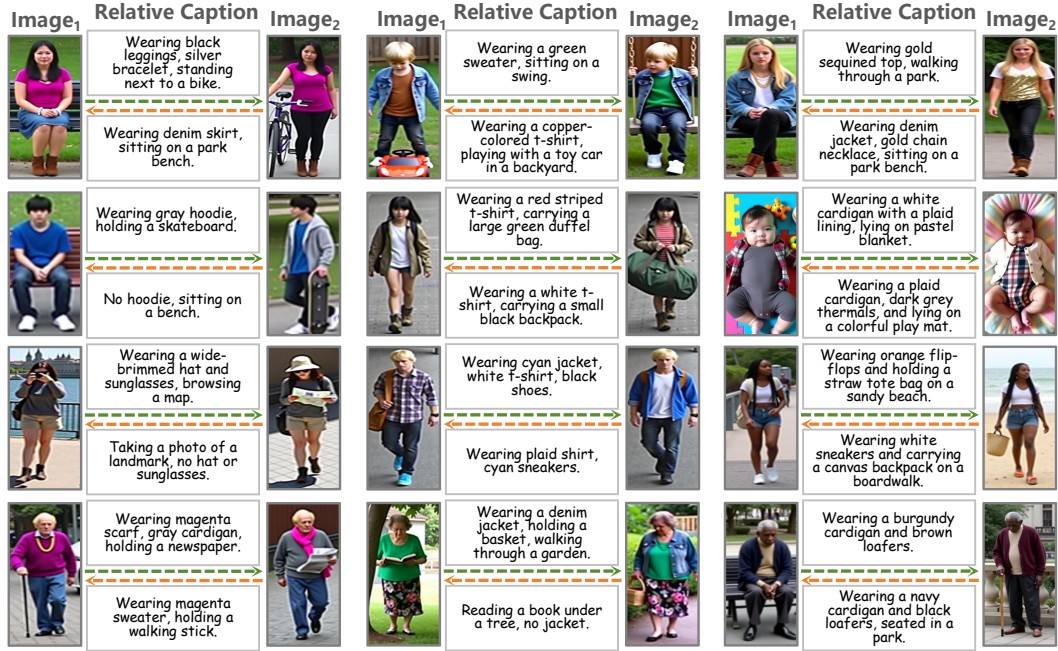

Figure S10: Representative examples of samples from the SynCPR dataset.

In total, 4,370 distinct words appear across all sentences, further highlighting the diversity of the SynCPR dataset.

The samples from SynCPR dataset are visualized in Figure S10. Thanks to our diversified textual generation strategy, realism-oriented fine-tuning of generative models, and rigorous data filtering

mechanisms, the SynCPR dataset ensures high quality, realism, and diversity of person images. By leveraging varied image generation prompts and the zero-shot generation capability of generative models, the SynCPR dataset encompasses rich scenarios, broad age coverage, diverse image clarity, varied attire and states of individuals, and comprehensive ethnic representation. Furthermore, the gender ratio in the generated dataset is highly balanced, with males accounting for 51.2%. Although SynCPR is entirely synthetic, its comprehensiveness significantly surpasses other manually annotated datasets within the person retrieval domain.

### B.3 CUHK-PEDES Dataset

CUHK-PEDES [4] is a widely used benchmark for text-to-person retrieval, comprising 40,206 pedestrian images and 80,412 corresponding textual descriptions annotated across 13,003 unique identities. The dataset is divided into three subsets: a training set containing 34,054 images and 68,108 descriptions for 11,003 identities, a validation set with 3,078 images and 6,158 descriptions for 1,000 identities, and a test set comprising 3,074 images and 6,156 descriptions for a separate set of 1,000 identities. Each image is paired with two independent human-written descriptions, and the average length of the descriptions exceeds 23 words.

## C  Additional Experiments and Results

### C.1  Additional Implementation Details.

All experiments are conducted on two NVIDIA H800 GPUs. For training the generative models, we select FLUX.1-dev [8] as the base model and apply LoRA with a rank of $r = 64$ specifically to the cross-attention layers. The training is performed with a per-GPU batch size of 1, utilizing gradient accumulation over eight steps, and optimized using AdamW [88] with an initial learning rate of $1 \times 10^{-5}$, 10-step warm-up, and weight decay of 0.01, for a total of 20,000 steps. The training employs bfloat16 mixed precision. Input person images from CUHK-PEDES are resized to $192 \times 384$ during training. During inference, each prompt dynamically adjusts LoRA strength, generating ten paired sub-images of size $400 \times 400$ via a 25-step beta noise reduction. Each generated image is first centrally cropped horizontally into two separate images, then each resulting sub-image is further centrally cropped to form a pair of person images sized $192 \times 384$.

For the Fine-grained Adaptive Feature Alignment (FAFA) framework, we use BLIP-2 [38] and a frozen ViT-G/14 [89] with an input resolution of 224 pixels. Input images undergo random horizontal flipping, random cropping with padding, and random erasing, followed by scaling the longer side to 224 pixels while preserving the aspect ratio. The images are then symmetrically padded horizontally to a final size of $224 \times 224$ before being input into FAFA. The model is trained on the SynCPR dataset using a single NVIDIA H800 GPU with a batch size of 256 for 10 epochs. Optimization is performed using AdamW [88] with an initial learning rate of $2 \times 10^{-6}$. In FAFA, the soft label strength parameter $\alpha$ is set to 0.5, the number of selected fine-grained features $k$ is set to 6, and the temperature parameter $\tau$ is 0.02. The margin parameter $m$ in the feature difference loss $\mathcal{L}_{fd}$ is 0.5. The loss balancing hyperparameters are set as $\lambda_1 = 1$ and $\lambda_2 = 0.5$. Additionally, for subsequent training from scratch on the Composed Image Retrieval (CIR) dataset CIRR [6], the model is trained for 50 epochs with an initial learning rate of $1 \times 10^{-5}$, while all other settings remain consistent.

In our experiments on CIRR, Rank-K serves as the primary metric, measuring the likelihood of finding the target image within the top-K retrieved candidates. For CIRR, we additionally report $Rank_s$-K on visually similar subsets, with overall performance summarized as $Avg. = \frac{\text{Rank-5}+\text{Rank}_s\text{-1}}{2}$.

### C.2  Additional Quantitative Results

#### C.2.1  FAFA for Composed Image Retrieval

The CPR task can be viewed as a more constrained and finer-grained variant of the CIR task, involving stricter alignment requirements between the reference and target images. Consequently, the FAFA framework, originally designed for CPR, can naturally be applied to CIR scenarios. We thus conduct experiments on CIRR, the most representative dataset within the CIR domain. The results, summarized in Table R1, demonstrate that our FAFA framework achieves comprehensive state-of-the-art performance with significant advantages, even in the context of CIR. Specifically,

Table S3: Performance comparison with existing supervised CIR methods on CIRR dataset only. The best results are marked in **bold**, and the second-best results are underlined. † indicates that the method is pretrained on its own constructed triplet dataset.

| Method | Ref. | Rank-K | | | $Rank_s$-K | | Avg. |
|---|---|---|---|---|---|---|---|
| | | K=1 | K=5 | K=10 | K=1 | K=3 | |
| TIRG [90] | CVPR19 | 14.61 | 48.37 | 64.08 | 22.67 | 65.14 | 35.52 |
| CIRPLANT [6] | ICCV21 | 19.55 | 52.55 | 68.39 | 39.20 | 79.49 | 45.88 |
| ARTEMIS [41] | ICLR22 | 16.96 | 46.10 | 61.31 | 39.99 | 75.67 | 43.05 |
| CLIP4CIR [42] | CVPR22 | 38.53 | 69.98 | 81.86 | 68.19 | 94.17 | 69.09 |
| TG-CIR [91] | MM23 | 45.25 | 78.29 | 87.16 | 72.84 | 95.13 | 75.57 |
| BLIP4CIR+Bi [43] | WACV24 | 40.15 | 73.08 | 83.88 | 72.10 | 95.93 | 72.59 |
| CASE† [92] | AAAI24 | 48.68 | 79.98 | 88.51 | 76.39 | 95.86 | 78.19 |
| CoVR-BLIP† [86] | AAAI24 | 49.69 | 78.60 | 86.77 | 75.01 | 93.16 | 76.81 |
| CompoDiff† [93] | TMLR24 | 32.39 | 57.61 | 77.25 | 67.88 | 94.07 | 62.75 |
| CaLa [47] | SIGIR24 | 49.11 | 81.21 | 89.59 | 76.27 | 96.46 | 78.74 |
| SPRC [48] | ICLR24 | 51.96 | 82.12 | 89.74 | 80.65 | 96.60 | 81.39 |
| **FAFA (Ours)** | - | **54.48** | **84.07** | **91.48** | **81.05** | **97.11** | **82.56** |

*Prompt1 ( $T_{I_q}$ ): A boy with blonde hair is wearing a khaki parka with a fur-lined hood, paired with dark blue jeans and brown hiking boots. He is walking in a snowy forest.*
*Prompt2 ( $T_{I_t}$ ): A boy with blonde hair is wearing a khaki parka with a fur-lined hood, but this time it's paired with a red scarf, black pants, and black snow boots. He is building a snowman.*

*Prompt1 ( $T_{I_q}$ ): A young adult with blonde hair is wearing a jade green tank top, denim shorts, and brown sandals. She is lounging on a beach chair under a parasol.*
*Prompt2 ( $T_{I_t}$ ): A young adult with blonde hair is wearing a jade green scarf over a white T-shirt, denim shorts, and brown sandals. She is walking along a path.*

Figure S11: Person image generation results under different LoRA strengths.

FAFA outperforms the second-ranked method SPRC, which also utilizes BLIP-2 as its backbone, by 2.51% in Rank-1 accuracy on the CIRR dataset. When compared with CaLa, another BLIP-2-based method, FAFA achieves an even more notable improvement, surpassing it by 5.37% in Rank-1.

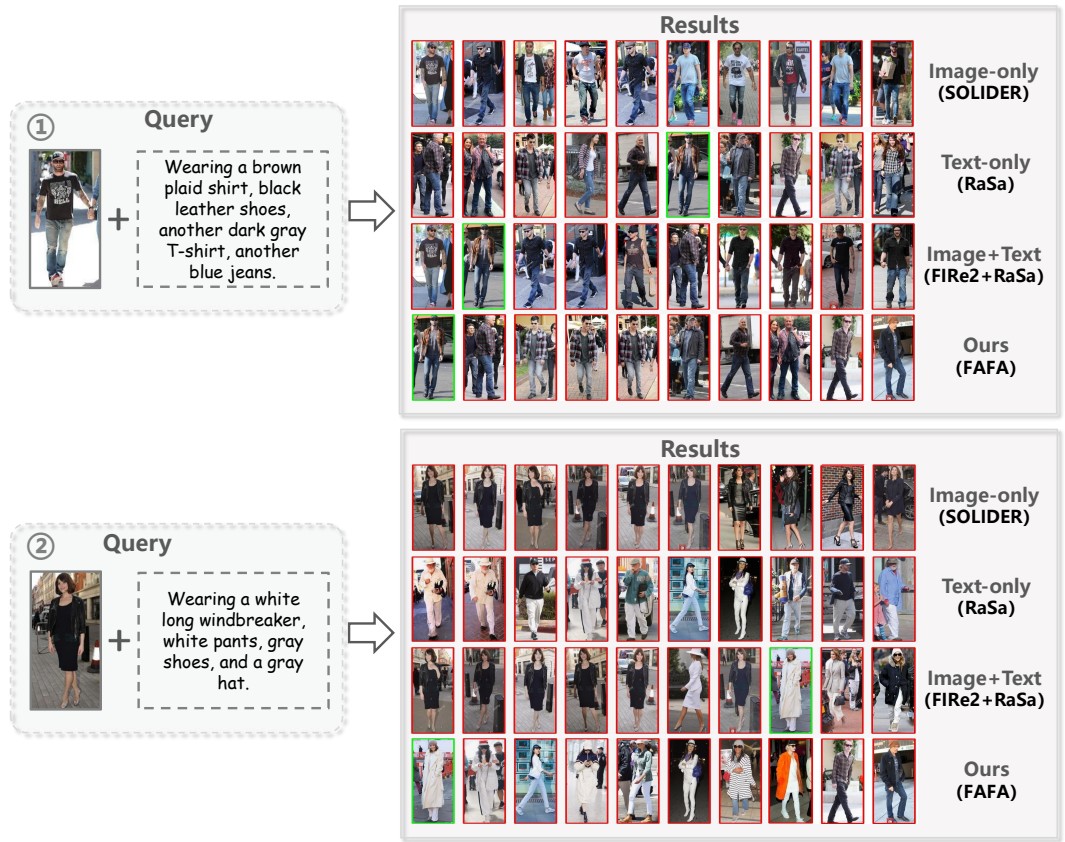

Figure S12: Comparative visualization of Top-10 retrieval results across different methods on the ITCPR dataset.

## C.3 Additional Qualitative Results

### C.3.1 Effects of Different LoRA Strengths on Person Image Generation

During the person image generation process, in order to achieve more comprehensive and realistic styles, multiple groups of images are generated for each textual prompt by dynamically adjusting the LoRA strength $\beta \in (0, 1]$. Figure S11 illustrates the effects of different LoRA strengths on generated person images, clearly demonstrating that the same textual input combined with varying values of $\beta$ can yield distinct image styles. Specifically, when $\beta = 0$, the pre-trained generative model is employed directly, resulting in high-quality images but with noticeable discrepancies from real-world styles. As the $\beta$ value increases gradually, the realism of the generated images correspondingly improves, ultimately reaching a style closely aligned with that of real-world person retrieval datasets at $\beta = 1$.

### C.3.2 Visualization of Results from Different Methods

Figure S12 presents two illustrative examples of the Top-10 retrieval results obtained by various representative methods from Table 1 on the ITCPR dataset. It is evident that the *Image-only* retrieval method yields the poorest performance, primarily because it tends to retrieve images with visually similar pixel distributions. Given that the dataset contains numerous instances involving clothing changes, this leads to suboptimal performance. *Text-only* retrieval also falls short of expectations, as most annotations in the dataset provide brief descriptions of clothing differences between the reference and target images, while the retrieval database includes many images with similar clothing. Combining both modalities typically retrieves the target image within the Top-10 results; however, its inability to dynamically complement multimodal query information leads to scenarios where an excessively high match in one modality adversely affects the final retrieval results. For instance, in

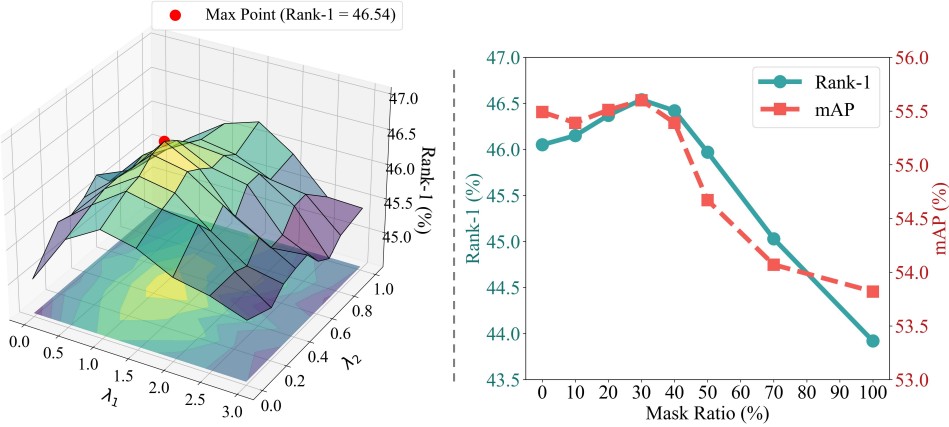

Figure S13: **Left**: Variations in FAFA's Rank-1 performance under different balancing weights of auxiliary loss terms. **Right**: Relationship between FAFA's performance and the feature mask ratio in $\mathcal{L}_{mfr}$.

Example ② of Figure S12, the high visual similarity causes the *Image + Text* method's most confident retrieval results to closely resemble those of the *Image-only* method. In contrast, our proposed FAFA method dynamically extracts complementary information from multimodal queries, consistently identifying the target person's image among the top-ranked retrieval results.

## C.4    Additional Ablation Study

### C.4.1    Balancing Weights of Auxiliary Loss Terms

To fully leverage the synergistic effects of the proposed loss functions, extensive experiments on balancing the weights of FAFA losses are conducted. As illustrated in Figure S13, when fixing the value of $\lambda_2$, the retrieval performance of FAFA initially rises and subsequently declines with increasing $\lambda_1$, achieving its highest performance at $\lambda_1 = 1$. Similarly, fixing $\lambda_1$ and varying $\lambda_2$ reveals the same trend, ultimately attaining the optimal performance at $\lambda_1 = 1$ and $\lambda_2 = 0.5$, which corresponds to our final selected configuration.

### C.4.2    Feature Masking Ratio in $\mathcal{L}_{mfr}$

As demonstrated in Figure S13, the optimal performance of FAFA in the $\mathcal{L}_{mfr}$ setting is achieved when the feature masking ratio is set to 30%. When the masking ratio is set to 0, it is equivalent to disabling the $\mathcal{L}_{mfr}$ loss. As the masking ratio increases, performance first improves and then declines. When the masking ratio exceeds 50%, the complexity of masked feature reasoning becomes excessively high, resulting in elevated loss values that negatively impact overall training stability, thereby diminishing the effectiveness of the $\mathcal{L}_{mfr}$ component.

