# OpenReview forum: "Automatic Synthetic Data and Fine-grained Adaptive Feature Alignment for Composed Person Retrieval"
_NeurIPS.cc/2025/Conference — NeurIPS 2025 poster_

### Official Review · Reviewer_qiB6 · 2025-06-29

**Clarity:** 4
**Significance:** 2
**Originality:** 2
**Rating:** 4
**Confidence:** 5

**Summary:**

This paper focuses on the topic of person retrieval. The author proposes **a new task** different from pure image retrieval or text retrieval, named composed person retrieval. In simple terms, this task uses an image and a text sentence describing changed clothing content as a combined query to search for images in the gallery, which is essentially clothing-changing REID. As a contribution to this new task, the author constructs **a pipeline for large-scale training data automatic synthesis**, thereby obtaining the SynCPR training dataset. The author also proposes **a new training framework** conducive to fine-grained alignment, named FIFA. In addition, the author constructs **a manually annotated test set** named ITCPR to more comprehensively and fairly evaluate the model's performance. The overall workload of this paper is relatively substantial.

**Questions:**

Here are some questions that interest me, and I would be grateful if the authors could provide answers.

1. How is the identity consistency ensured in your data generation process? Limited by the capabilities of existing text-to-image models, can the method shown in Figure 3 truly ensure ID consistency? Is there any quantitative experiment to prove this?

2. You only used the training set of CUHK-PEDES to fine-tune FLUX, but there are other datasets in the text person reid field, such as ICFG-PEDES and UFine6926. Should you attempt to fine-tune these datasets as well, perhaps to better enhance the real-world generalization ability of your generation model?

3. The experiments in Table 1 seem unconvincing in demonstrating the effectiveness of your FAFA. You only fine-tuned the CIR method using your SynCPR dataset, while other methods are basically cross-domain experiments, making this comparison unfair. Additionally, the CIR method lacks designs for person-relevant aspects such as identity matching, so it is reasonable that your method outperforms CIR. Considering that TPR methods essentially support composed retrieval as well, you should focus on comparing with TPR methods by fine-tuning these TPR methods on your SynCPR dataset first, and then conduct a fair comparison.

4. Conducting cross-domain and cross-dataset experiments can better demonstrate the effectiveness and generalizability of your method, rather than only evaluating its performance on the test set you proposed.

Overall, this paper has made substantial contributions, but there are still some shortcomings. If you can address the questions raised above, I am willing to raise my score.

**Ethical Concerns:**

["NO or VERY MINOR ethics concerns only"]

**Final Justification:**

The contributions of this paper are sufficient, and the authors' detailed responses have addressed most of my questions. It is hoped that the authors will include the discussions in the final version.

**Limitations:**

The authors have barely discussed the limitations, especially the negative social impacts of their work. The authors should seriously and thoroughly discuss the social impacts of this technology in terms of privacy, and propose some control measures to prevent the improper use of this technology.

**Paper Formatting Concerns:**

There are no formatting issues in this paper.

**Quality:**

3

**Strengths And Weaknesses:**

Strengths:
1. This paper proposes a new task named Composed Person Retrieval. The task uses both images and text as queries to search for images in the gallery. Although this task has been proposed in the general domain (composed image retrieval), such in-depth task definition and exploration of solutions are the first in the person retrieval field.
2. This paper makes significant overall contributions, proposing a new task, a new training data synthesis strategy, a new training framework, and a new real-world test set.
3. The paper is well-written, fluent and easy to understand, with clear explanations of details, making it easy for readers to follow.

Weaknesses:
1. The experiments are somewhat insufficient. The authors mainly conducted two types of experiments: one is the performance comparison with previous methods, and the other is the ablation experiments to verify the effectiveness of each proposed component. In fact, I believe the focus of the experiments should lie in verifying the rationality and uniqueness of this new task. The authors should have a separate section for comprehensive analysis and experimental verification of this new task. For instance, what are the essential differences between this task and existing person retrieval tasks or traditional composed image retrieval? How much improvement can image-text combination queries bring? Etc. Conducting some qualitative and quantitative experiments to verify the practical application rationality of this task is, in my view, the most core and important part.

2. The research object of this paper is persons, and it aims to utilize their biological characteristics for large-scale searches, which inevitably raises significant privacy concerns and negative social impacts. However, instead of discussing these issues or proposing reasonable control strategies against technical abuse, the authors arbitrarily claim that this research "does not pose any obvious negative societal impacts."

3. This paper seems to lack comparisons with some related works. As far as I know, in the field of person re-identification, there have been several studies on image-text combined retrieval, but the authors do not discuss or compare these methods. For example, relevant papers such as InstructReID and UNIReID.

---

> ### Author Rebuttal · Authors · 2025-07-30
>
> Thank you very much for your valuable and insightful feedback. We carefully address each of your concerns as follows.
>
> ## Responses to Weaknesses:
>
> **A1.1: Insufficient Experimental Validation**
>
> Considering the space limitations, we arrange more experimental details and supplementary results in the Appendix.
> It is worth pointing out that Table 1 is not just a performance comparison, but a comprehensive summary that addresses your multiple key concerns.
> Firstly, regarding the rationality and uniqueness of the new Composed Person Retrieval (CPR) task, the Introduction highlights its practical significance: in real-world scenarios, visual and textual information are often both accessible. For example, when elderly individuals or children go missing, which occurs frequently, we usually have previous photos as well as verbal descriptions of their current appearance. Therefore, CPR is specifically designed to leverage such complementary multimodal information for person retrieval in realistic search scenarios. Please refer to the response for Reviewer AeXC (A1.2) for more potential application examples.
>
> Secondly, regarding empirical validation, the first three experimental settings in Table 1 demonstrate that relying solely on either visual or textual modalities (as is common in existing person retrieval methods) is ineffective for CPR. In contrast, a simple combination of image and text (the 4th setting in Table 1) substantially improves retrieval performance, validating both our CPR task formulation and the relevance of the ITCPR annotations. Further improvements through our proposed FAFA framework and training on SynCPR show the importance of effectively extracting complementary multimodal information. These results also indicate that the proposed modules are indispensable.
>
> We will provide more detailed analysis and interpretation in the revised manuscript. Additionally, we have added new experiments to strengthen the sufficiency. Please refer to the responses for Reviewer iMvu (A1.1 and A2.2), Reviewer ZBdY (A3.1), and Reviewer AeXC (A1.1).
>
> **A1.2: Privacy Concerns and Negative Social Impacts**
>
> Thank you for raising these ethical concerns. It should be noted that: (1) Our SynCPR dataset is completely synthetic, with all person images depicting entirely fictional individuals, without involving any real identities or non-public materials. (2) Our ITCPR test set is constructed from publicly available datasets (Celeb-reID, PRCC, LAST), explicitly permitting secondary editing and academic usage. Therefore, there are no risks in terms of privacy issues or negative social impacts.
> In the revised version, we will modify and clarify the descriptions that may lead to potential risks, to reflect the social responsibility and core value of "Tech for Good".
>
> **A1.3: Insufficient Comparisons with Related Works**
>
> Thank you for suggesting relevant works, and we will add discussions of these methods in our revised Related Work section. However, it should be noted that these methods are not quite suitable for the CPR task we propose. For instance, the InstructReID model typically uses fixed textual instructions for most individuals. When it receives flexible textual instructions combined with images, the textual content tends to duplicate rather than complement the image information. Essentially, this belongs to modality fusion rather than complementary multimodal input. Consequently, InstructReID is not suitable for CPR scenarios that require complementary rather than redundant information.
>
> Moreover, the UNIReID model is limited to forensic scenarios involving sketch-plus-text retrieval and has not publicly released its trained checkpoints, limiting direct comparisons. We thus focus on experimentally comparing InstructReID's inference performance on our CPR task:
>
> | Method | Test Data | Rank-1 | Rank-5 | Rank-10 | mAP  |
> | - | - | - | - | - | - |
> | InstructReID | Text-Only   | 27.02  | 47.18  | 56.90  | 36.79 |
> | InstructReID | Combination | 26.29  | 46.50  | 56.18  | 36.21 |
> | FAFA (Ours)  | Combination | 46.54  | 66.21  | 73.12  | 55.60 |
>
> Clearly, the InstructReID model fails to effectively utilize complementary multimodal information for CPR. The performance of its combined input is even lower than that of single-modal textual input, highlighting FAFA's superiority for this specific task.
>
> ## Responses to Questions:
>
> **A2.1: Identity Consistency in Data Generation (Quantitative Validation)**
>
> As described in Section 3.1, our data synthesis pipeline ensures identity consistency by generating two related sub-images simultaneously within a single image. Qualitative evidence is provided by Figure 3 in the manuscript and Figure S10 in the Appendix. Moreover, we agree that quantitative verification is necessary, and thus we conduct an additional experiment. Specifically, we randomly select 10 000 triplets (5 000 quadruples) from SynCPR and compare two generation strategies—individual image generation and simultaneous paired generation—using identical prompts and the same fine-tuned Flux.1 model. To ensure a fair comparison, generated images that do not meet the multimodal filtering criteria (excluding identity consistency) are regenerated.
>
> We utilize the FIRe2 [1] model (pre-trained on LTCC [2]) to compute an **Identity Recall** metric. For each of the 10 000 images (5 000 identities), we retrieve the top-10 matches from the remaining images; if the same identity appears in these matches, it is regarded as a successful recall. Identity Recall is the percentage of successful recalls. Retrieval performance (Rank-1, Rank-5, Rank-10, mAP) using FAFA trained on these datasets is also reported:
>
> | Method | Training Data | Rank-1 | Rank-5 | Rank-10 | mAP | Identity Recall (%) |
> | - | -| - | - | - | - | - |
> | FAFA  | Individual generation | 37.51  | 57.81  | 67.30 | 47.11 | 2.17 |
> | FAFA | Simultaneous generation | 41.51  | 62.40  | 69.71 | 52.03 | 72.35  |
>
> These results clearly demonstrate the effectiveness of our simultaneous-generation approach in ensuring identity consistency. This quantitative analysis will be included in the revised manuscript.
>
> [1] Exploring fine-grained representation and recomposition for cloth-changing person re-identification. TIFS, 2024
>
> [2] Long-term cloth-changing person re-identification. ACCV, 2020
>
> **A2.2: Choice of Dataset for Fine-tuning**
>
> Regarding the concern about whether all datasets should be used: our goal is to propose a feasible solution for the new CPR task, which serves as an important supplement to existing person-retrieval technologies such as text-based person retrieval (TPR), image-based person retrieval (IPR), and cloth-changing Re-ID. Therefore, UFine6926, which primarily collects images from online videos, is excluded. Among the three classic TPR datasets (CUHK-PEDES, ICFG-PEDES, RSTPReid), only CUHK-PEDES gathers images from five surveillance datasets, providing broader scene diversity. The other two datasets source their images from MSMT17, which mainly contains a single parking-lot scenario. To prevent overly homogeneous scene distributions from negatively affecting generation diversity, we choose CUHK-PEDES for fine-tuning. We will briefly add this reasoning to the revised manuscript.
>
> **A2.3: Fine-tuning TPR Methods**
>
> We clarify a misunderstanding: all compared methods, including our own, are independently trained without referencing the ITCPR test-set distribution, meaning that every experiment constitutes a cross-domain test. Following your suggestion, we fine-tune two representative TPR methods (IRRA and RDE) on our SynCPR dataset for a fair comparison. Reference images and textual descriptions are combined into a global query feature by summing features from dual encoders, while target images are encoded into global features separately. We train these methods under two settings: training from scratch and fine-tuning from pre-trained checkpoints on TPR datasets. The results are as follows:
>
> | Method | Pretraining Data | Rank-1 | Rank-5 | Rank-10 | mAP   |
> | - | - | - | - | - | - |
> | IRRA   | – | 27.56  | 47.41  | 56.95   | 37.23 |
> | IRRA   | CUHK-PEDES | 35.33  | 55.00  | 63.81   | 44.86 |
> | RDE    | – | 30.43  | 52.68  | 61.26   | 40.85 |
> | RDE    | CUHK-PEDES       | 36.92  | 59.17  | 67.03   | 47.21 |
> | FAFA   | – | 46.54  | 66.21  | 73.12   | 55.60 |
>
> These results show that pretraining on TPR datasets improves retrieval performance. Nevertheless, TPR methods remain far inferior to our FAFA and the fine-tuned Composed Image Retrieval (CIR) approaches (Table 1). CIR methods leverage attention-based interactions between reference images and relative textual descriptions, whereas TPR methods encode these independently, limiting their ability to exploit complementary multimodal information effectively.
>
> **A2.4: Cross-domain and Cross-dataset Experiments**
>
> As explained in A2.3, all of our current experiments are conducted under cross-domain settings. To the best of our knowledge, CPR is a newly proposed task, and there are presently no alternative datasets available for broader testing. Nevertheless, in Table S3 of the Appendix we additionally compare FAFA on a CIR benchmark that also uses triplet supervision; FAFA still achieves state-of-the-art performance, further demonstrating the effectiveness and robustness of our method.  We welcome the Reviewer to provide datasets suitable for cross-dataset validation for the additional experiments.
>
> ## Response to Limitations:
>
> **A3:** We will significantly expand our discussion of limitations and potential negative social impacts as described in response A1.2. Additionally, introducing multiple reference images or textual descriptions to improve practicality, as well as deploying the CPR model under resource-constrained conditions, remains to be explored.

---

> > ### Comment · Reviewer_qiB6 · 2025-08-02
> >
> > Thank you for your detailed response. After reading it, I still have the following questions, and I hope you can answer them:
> >
> > 1. About "Identity Consistency in Data Generation" Experimental Setup. Could you please provide a more detailed explanation of how your Rank and Identity Recall metrics are calculated? Why is there not a significant difference in the Rank metric between the two methods, while the Identity Recall metric shows a large discrepancy? I currently don't have a clear understanding of your experimental setup for this section.
> >
> > 2. About "Choice of Dataset for Fine-tuning". Why was UFine6926 not selected for fine-tuning? You simply stated that the images in this dataset are from online videos, which I find unconvincing. Why are online video sources unsuitable? Doesn't the diversity of scenarios in online videos offer value to the task? Considering that UFine6926 contains more fine-grained text-image data with better matching than other datasets, it might be more beneficial for fine-tuning. I hope you can provide a more detailed discussion on this part.
> >
> > 3. Can your method still perform retrieval when either the text or image modality is missing? In other words, does your method support both single-modal retrieval and combined-modal retrieval? Given that modality loss is a common phenomenon in real-world applications, I hope you can discuss this in more detail.
> >
> > I am very interested in these questions and look forward to your answers.

---

> > > ### Author Response · Authors · 2025-08-02
> > >
> > > Thank you for your insightful comments. Our responses are as follows:
> > >
> > > **1. Identity Consistency in Data Generation (Experimental Setup)**
> > >
> > > The Rank and mAP metrics reported are evaluated on the ITCPR test set using our FAFA model trained on the corresponding data. To obtain good performance in terms of Rank and mAP, it is required to achieve semantic alignment within each image-text "triplet" (reference image, text description, target image). Additionally, the strict identity consistency in synthesized images further improves retrieval performance, due to the stronger constraints on simultaneously generated images. Our experiments confirm this: simultaneously generating identity-consistent image pairs significantly increases the Rank-1 (from 37.51% to 41.51%) and the mAP (from 47.11% to 52.03%).
> > >
> > > To quantitatively validate the identity consistency, we introduce the independent **Identity Recall** metric:
> > >
> > > (1) We generate 10,000 images for 5,000 identities (two images per identity) using simultaneous-generation and individual-generation strategies. An independent cloth-changing Re-ID model (FIRe2) pretrained on the LTCC dataset is employed for retrieval.
> > > (2) Each of the 10,000 images serves as a query, retrieving the top 10 matches from the remaining 9,999 images.
> > > (3) Identity Recall measures the percentage of queries for which the paired image (same identity) appears among the top 10 matches.
> > >
> > > Results show that Identity Recall is significantly higher for simultaneous-generation (72.35%) compared to individual-generation (2.17%), clearly demonstrating our strategy’s advantage in identity consistency.
> > >
> > > **2. Choice of Dataset for Fine-tuning (UFine6926)**
> > >
> > > Our proposed CPR task aims to effectively integrate image and text modalities for realistic scenarios and complement existing methods (TPR, IPR, and cloth-changing Re-ID). The selection of datasets for fine-tuning gives priority to the representativeness and the generalization capability in the real world. Through comparison, we notice that UFine6926 is built from web videos and has the extremely high image quality (e.g., “the third button on the blue striped shirt” can be easily identified). However, this differs significantly from the vast majority of existing pedestrian datasets (such as Market-1501, DukeMTMC, CUHK03 etc.) and the pedestrian images detected from real scenes (a large number of which are of low quality). Therefore, considering the obvious domain differences, fine-tuning based on UFine6926 is very likely to weaken the model's ability to handle low-quality scenes in reality.
> > >
> > > Additionally, the diversity of synthesized data primarily depends on the generative model’s ability to capture realistic person image styles, rather than relying solely on original dataset diversity. Fine-tuning can anchor the generative model’s visual style without sacrificing zero-shot generalization, and the diversity can be achieved through varied textual prompts. Figures S10 and S11 illustrate diverse realistic scenarios generated by the model, many of which are absent from the original training data.
> > >
> > > Specifically, Figure S11 further highlights the advantage of fine-tuning on CUHK-PEDES, which is a dataset reflecting real-world scenarios. After fine-tuning with CUHK-PEDES images (intensity β=1), we can flexibly simulate various realistic or photographic image styles during inference by adjusting the intensity parameter β within [0,1]. Conversely, if UFine6926 had been initially selected, achievable styles would be greatly limited (approximately β∈[0,0.7]), thus restricting stylistic diversity in our final SynCPR dataset.
> > >
> > >
> > > ### 3. Retrieval under Missing Modalities
> > >
> > > To address modality deficiency, we adjust FAFA’s training strategy by explicitly including modality-deficient scenarios:
> > >
> > > * Complete modality inputs (image + text): 70%
> > > * Text-modality deficiency (blank text): 15%
> > > * Image-modality deficiency (purely black images): 15%
> > >
> > > We apply the same conditions during inference on ITCPR and obtain these results:
> > >
> > > | Method | Test Data | Rank-1 | Rank-5 | Rank-10 | mAP |
> > > | - | - | - | - | - | - |
> > > | FAFA | Complete | 45.87 | 65.53 | 72.52 | 54.90 |
> > > | SOLIDER | Image-only | 8.45 | 18.48 | 23.89 | 13.74 |
> > > | FIRe2 | Image-only | 10.76 | 22.84 | 29.29 | 17.00 |
> > > | FAFA | Image-only | 12.08 | 24.43 | 33.65 | 19.38 |
> > > | RDE | Text-only | 26.43 | 47.41 | 56.45 | 36.43 |
> > > | RaSa | Text-only | 28.02 | 49.23 | 57.77 | 38.04 |
> > > | FAFA | Text-only | 29.61 | 51.59 | 60.22 | 39.84 |
> > >
> > > We obtain two conclusions:
> > >
> > > First, explicitly introducing modality-deficient scenarios only slightly reduces FAFA’s performance with complete inputs (compared to Table 1), demonstrating the robustness of our training strategy.
> > >
> > > Second, under single-modality conditions, FAFA notably outperforms existing specialized methods for image retrieval (SOLIDER, FIRe2) and text retrieval (RDE, RaSa). FAFA thus supports both combined- and single-modality retrieval effectively, showing strong generalization.

---

> > > > ### Comment · Reviewer_qiB6 · 2025-08-03
> > > >
> > > > Thank you for your detailed reply. Most of my concerns have been addressed. I hope you can include the above discussions in the final version of your paper (for example, in the section "Choice of Dataset for Fine-tuning"). If there is not enough space, please at least include them in the Appendix. I decide to raise the score to 4 (Borderline accept). Good luck to you.

---

### Official Review · Reviewer_AeXC · 2025-06-29

**Clarity:** 3
**Significance:** 2
**Originality:** 3
**Rating:** 5
**Confidence:** 4

**Summary:**

This paper explores Composed Person Retrieval (CPR), a new task that utilizes a compositional query, comprising a reference image and a textual description of modifications, to retrieve (re-identify) a target person.

The paper presents two main contributions: the construction of the SynCPR dataset and the Fine-grained Adaptive Feature Alignment (FAFA) framework.
To generate this high-quality synthetic dataset, a novel data synthesis pipeline is proposed, which uses a Large Language Model (Qwen2.5-70B) to generate textual quadruples, a fine-tuned generative model (Flux.1) to create identity-consistent images, and a Multimodal Large Language Model (Qwen2.5-VL) to filter for high-quality data.
Based on this dataset, the FAFA framework is designed to improve fine-grained retrieval accuracy using three key components: Fine-grained Dynamic Alignment (FDA), Feature Diversity (FD) supervision, and bidirectional Masked Feature Reasoning (MFR).

Experimental results on a newly annotated benchmark, ITCPR, confirm the effectiveness of both the SynCPR dataset and the FAFA framework when compared to existing methods.

**Questions:**

Based on Weaknesses that I mentioned above, I'd like to suggest:

1. Please justify using more fine-grained similarity (Eq.(3)) for evaluation. Compared to using the average pooled target image feature $\bar{f_t}$ and other existing methods, how much more computational cost does ${f^i_t}^{N_t}_{i=1}$ require, and what is the performance difference?

2. The justification for the practical usefulness of the proposed CPR task should be strengthened. The paper would be more compelling with a discussion of more diverse and concrete application scenarios to better motivate the task's significance.

3. Please provide more comprehensive intuitions and additional analysis behind Feature Diversity (FD) and Masked Feature Reasoning (MFR).

**Ethical Concerns:**

["Major Concern: Data quality and representativeness", "Major Concern: Human rights (including surveillance)"]

**Final Justification:**

I appreciate the authors' additional experimental results and discussions. All of the major concerns I raised have been addressed during the rebuttal period. Taking this into account, I'd like to raise my score to 5.

**Limitations:**

I didn't see specific sections or sentences related to this and I hope this part will be further revised.

**Paper Formatting Concerns:**

None.

**Quality:**

3

**Strengths And Weaknesses:**

Overall, I think the paper presented sufficient contributions, but I have some conerns.

**Strengths**
1. The synthetic dataset generation pipeline is well-designed and its effectiveness in CPR tasks has been experimentally verified in Table 1. Specifically, the process of generating consistent identity image pairs for a fine-grained person retrieval task is novel and effective. Also, the open-sourcing of related codes and datasets is appreciated.

2. The ITCPR benchmark for the CPR task is also valuable, as its construction involved substantial manual annotation effort and noise-handling processes.

3. The training pipeline and intuition of the FAFA framework seem reasonable, and it demonstrated strong performance in the experiments.

**Weaknesses**

1. As stated in L219-L222, the proposed framework seems to use multiple feature vectors to represent database images. More specifically, it uses a feature similarity defined in Eq(3) during inference. However, to my knowledge, most existing CIR approaches utilize a single feature vector during the evaluation stage. This implies that the proposed method requires more storage space and computational cost during the search stage. Please justify this trade-off.

2. The usefulness of the proposed CPR task is questionable. General composed image retrieval has clear usage examples, such as product searches that reflect a user's intention on an e-commerce platform. However, what about composed person retrieval? Given that a primary application of person retrieval is in surveillance systems, it is questionable how useful CPR would be. I don't think this issue renders the paper's academic contributions meaningless, but its justification is a significant issue.

3. Based on Table 2, the effectiveness of Feature Diversity (FD) and Masked Feature Reasoning (MFR) seems marginal compared to that of FDA. It would be better to have additional intuition and analysis on these auxiliary losses. The current manuscript primarily discusses the numerical results. The paper would be strengthened by further clarifying the behavior and intuition of these losses through additional qualitative analysis or comparison.

---

> ### Author Rebuttal · Authors · 2025-07-30
>
> Thank you very much for your valuable feedback and positive assessment. We reply to each of your concerns below.
>
> ## Responses to Weaknesses/Questions:
>
> **A1.1: Efficiency and Computational Cost of Using Multiple Feature Vectors**
>
> During inference, existing Composed Image Retrieval (CIR) methods vary significantly in terms of the number of target-image tokens they use. For instance, among methods based on BLIP2 (CaLa and SPRC), CaLa only uses the first token as a global image feature, while SPRC uses all 32 tokens and selects the one with the highest similarity score as the final similarity. Regardless, the primary computational cost arises from feature extraction rather than feature storage or retrieval operations.
>
> In terms of storage, CaLa stores a single 768-dimensional vector per image, while both SPRC and our FAFA store 32 vectors (each with 256 dimensions). For a database containing 10,000 images stored in half-precision format, CaLa requires approximately 15.36 MB, while SPRC and FAFA require about 163.84 MB. Such an amount of storage is relatively easy to manage.
>
> We evaluate the inference efficiency on the entire ITCPR dataset using a single H800 GPU (the average of 10 runs, with the time unit being seconds):
>
> | Method | 2202 Query Feature Extraction | 20510 Candidate Feature Extraction | Retrieval |
> | - | - | - | - |
> | CaLa | 7.9549s | 48.7663s | 1.6892s |
> | SPRC  | 9.5462s  | 56.1453s  | 1.0406s |
> | FAFA (k=1)  | 7.1764s | 56.0928s | 1.0291s |
> | FAFA (k=6)  | 7.1778s | 56.0124s | 1.0337s |
> | FAFA (k=16) | 7.1598s | 56.0854s | 1.0445s |
>
> These results show that feature extraction dominates computational cost. The retrieval step itself (involving simple matrix operations) consumes minimal time. Specifically, our FAFA achieves the fastest query feature extraction as it involves only a Q-Former and a visual encoder, unlike SPRC and CaLa, which include additional text encoders. Although CaLa can extract candidate features more quickly, its retrieval speed is the slowest. Importantly, varying the parameter `k` in FAFA (the number of tokens used for similarity computation) has a negligible impact on retrieval time. Therefore, selecting `k` based on retrieval accuracy is optimal; the original Figure 5(b) reports retrieval performance for `k` values from 1 to 16, with `k = 6` achieving the best performance and thus chosen as our final setting.
>
> **A1.2: Practical Justification for the CPR Task**
>
> Thank you for raising this practical question. The Composed Person Retrieval (CPR) task has significant application value in multiple aspects:
>
> * **Missing person search**: Families usually provide a past photograph and verbal descriptions of the current appearance of the person involved. A CPR system integrating both image and textual descriptions can efficiently give the most relevant results from surveillance or social media images, etc. This situation happens frequently, especially with children and the elderly.
> * **Media editing and content management**: Video editors can efficiently retrieve specific clips using actor photographs combined with descriptions of desired scenes or states.
> * **Personalized fashion recommendation**: Users on e-commerce platforms can upload personal photos along with textual style descriptions, thus precisely retrieving matching outfits and enhancing shopping experiences.
>
> Therefore, CPR has broad applicability. We will incorporate this discussion in the revised manuscript.
>
> **A1.3: Intuitions and Analysis of Feature Diversity (FD) and Masked Feature Reasoning (MFR)**
>
> **Feature Diversity (FD)**
>
> *Intuition:*
> In the CPR task, fine-grained target features precisely capture subtle appearance differences associated with the compositional query. Without FD supervision, extracted features may become redundant, limiting the ability to distinguish delicate differences. FD supervision encourages the extracted features to be sufficiently distinctive and complementary, thus fully representing the rich details within person images.
>
> *Qualitative Analysis:*
> FD loss sets a maximum cosine similarity threshold `m`, pushing features apart in the semantic space. This helps form diverse representations and enhances sensitivity to subtle variations. Specifically, the model trained with FD (No.5 in Table 2) reduces the intra-group maximum cosine similarity of target features from 0.824 (No.4 in Table 2) to 0.713, indicating that it contains richer and more diverse information and is suitable for Fine-grained Dynamic Alignment (FDA).
>
> *Theoretical Analysis:*
> From the perspective of representation learning, FD acts similarly to a regularization mechanism, maximizing the semantic coverage and discriminability of the target feature space. From an information-theoretic viewpoint, FD reduces redundancy and uncertainty and increases entropy, thus enhancing representational efficiency and generalization, especially when the test distribution is unknown.
>
> **Masked Feature Reasoning (MFR)**
>
> *Intuition:*
> In the CPR task, the combination of the reference image and textual description inherently involves complementary information, as the text describes visual changes. MFR randomly masks parts of the combined representation to force the model to explicitly capture this complementarity, thus improving its capability to recover complete representations from partial information.
>
> *Qualitative Analysis:*
> MFR enhances the extraction of complementary information between reference images and text, thus improving alignment with target features. An intuitive demonstration method is to visualize the attention distributions when extracting query features and observe if textual emphasis and image regions become more complementary after adding MFR supervision. We will include this visualization in the revised manuscript (images cannot be inserted in the current reply). Additionally, the average cosine similarity between query features and FDA-derived target features increases from 0.481 (No.3 in Table 2) to 0.504 (No.5 in Table 2), indirectly confirming MFR's effectiveness in promoting complementary information extraction.
>
> *Theoretical Analysis:*
> Theoretically, MFR is similar to masked autoencoder mechanisms, improving the model's understanding of input features through a self-supervised masking constraint. Randomly masking feature dimensions can act as a self-supervised objective, promoting effective modeling of conditional probability distributions between different modalities. This implicit reconstruction objective increases the density and continuity of the cross-modal feature space and theoretically ensures that more robust and generalizable representations can be obtained.
>
> ## Response to Limitations:
>
> **A2:**
> Thanks for your forward-looking comments. A more detailed discussion regarding the limitations will be expanded in the revised version. Firstly, following mainstream person retrieval approaches, FAFA currently only supports taking one image and one textual description as input. However, a feasible approach for introducing multiple reference images or textual descriptions to further improve practicality is discussed in our response to Reviewer iMvu (A2.3). Additionally, the application of the CPR model under resource-constrained conditions also remains to be further addressed.
>
> ## Response to Ethical Concerns:
>
> **A3.1: Data Quality and Representativeness**
>
> Our SynCPR dataset is entirely generated through an automated pipeline, involving three stages: the generation of diverse textual quadruples, the synthesis of identity-consistent images, and rigorous multimodal quality filtering. Specifically, approximately 140,000 diverse textual quadruples are generated by Qwen2.5-70B; subsequently, identity-consistent image pairs are synthesized by the Flux.1 model fine-tuned via LoRA, with additional style perturbations to enhance diversity. Finally, a strict multi-criteria filtering approach retains 1.15 million high-quality triplets. These triplets cover extensive variations in clothing, age, scene, and so on, ensuring high diversity and representativeness (Supplementary Figure S10 provides an intuitive illustration of this point). Therefore, there are no risks arising from biases or distortions.
>
> **A3.2: Human Rights and Surveillance Concerns**
>
> It should be noted that: (1) Our SynCPR dataset is completely synthetic; all person images in it depict entirely fictional individuals, without involving any real identities or non-public materials. (2) Our ITCPR test set is constructed from publicly available datasets (Celeb-reID, PRCC, LAST), explicitly permitting secondary editing and academic usage. Therefore, there are no risks in terms of ethics or human rights.

---

> > ### Comment · Reviewer_AeXC · 2025-08-02
> >
> > Thank you for the authors' response and efforts. I still have a few questions regarding the FAFA framework:
> >
> > 1. I remain curious about the performance of the FAFA framework when using only the average pooled target image feature $\bar{f}_t$. Since most person re-identification and retrieval methods commonly utilize feature vectors of 768 or 2048 dimensions, the feature dimension used in FAFA (32 × 256) appears relatively large. Would it be possible to achieve reasonable performance using only the average pooled target feature as a global representation?
> >
> > 2. The FAFA framework employs input images of size 224 × 224, while the proposed SynCPR dataset provides images at 192 × 384 resolution. Could you clarify the rationale behind using inconsistent image resolutions? Additionally, how does FAFA perform when trained with the original 192 × 384 image resolution?

---

> > > ### Author Response · Authors · 2025-08-03
> > >
> > > Thank you for your two very valuable questions.
> > >
> > > **1. Retrieval Performance with Average-Pooled Target Features**
> > >
> > > We conduct additional experiments as follows: keeping the Feature Diversity (FD) and Masked Feature Reasoning (MFR) losses unchanged, we only adjust the parameter k in the Fine-grained Dynamic Alignment (FDA) loss to 32.Then, we perform average pooling on these 32 sub-features to obtain the average-pooled target image features. The results are shown below:
> > >
> > > | Method | Strategy (k) | Rank-1 | Rank-5 | Rank-10 | mAP |
> > > | - | - | - | - | - | - |
> > > | CaLa (SynCPR)| - | 39.33 | 60.85 | 68.66 | 49.29 |
> > > | SPRC (SynCPR)| - | 42.27 | 61.81 | 69.35 | 51.62 |
> > > | FAFA   | 1 | 44.35 | 64.63 | 71.54 | 54.24 |
> > > | FAFA   | 32 | 44.02 | 65.01 | 71.89 | 54.30 |
> > > | FAFA   | 6 (Ours) | 46.54 | 66.21 | 73.12 | 55.60 |
> > >
> > > These results indicate that compared with the case of k=1, using average-pooled features leads to a slight decrease in Rank-1 performance, while other metrics show minor advantages, resulting in comparable overall performance. However, since this setting turns off the dynamic feature selection function, each image only needs to store a single global feature, and the storage volume is significantly reduced, which makes it particularly suitable for large-scale retrieval databases. It is worth noting that even under this storage-efficient setup, FAFA can still achieve competitive performance and outperforms other baseline methods. Of course, when storage is not an issue, we still recommend the original configuration presented in our manuscript.
> > >
> > >
> > > **2. Rationale for Image Resolution Choices**
> > >
> > > Regarding the SynCPR dataset, we choose a 1:2 width-to-height aspect ratio for person images to ensure the individual occupies most of the frame, which is consistent with most Re-ID datasets. Specifically, the resolution of 192 × 384 is selected as a trade-off after taking two main factors into consideration:
> > > Firstly, by fine-tuning the generative model with real TPR datasets and performing inference at full strength (β=1), it is easy to generate lower-quality person images even at relatively higher resolutions.
> > > In addition, a diverse dataset also needs to contain clear person images, which means that excessively low resolutions are inappropriate. Therefore, the resolution of 192 × 384 is an appropriate balance, which enables the efficient synthesis of images with diverse styles through dynamic adjustment of the β parameter.
> > >
> > > For the FAFA retrieval model, its visual encoder (EVA-ViT-G [1]) in the pretrained BLIP-2 is initially trained on images with a size of 224 × 224. Due to the large number of parameters in the visual encoder, directly training it within FAFA will dramatically increase computational resources, and it will even exceed GPU memory limits on a single H800 GPU (retraining is very costly). Hence, we freeze the visual encoder and primarily rely on its pretrained capability to extract features. Although resizing the positional embeddings can make the visual encoder partially adapt to different input resolutions, we empirically test three input resolution strategies (keeping other settings identical):
> > >
> > > * Input resolution 192 × 384, resized positional embeddings.
> > > * Input resolution 224 × 224, directly resizing images.
> > > * Input resolution 224 × 224 (our final setting), resizing images by scaling the longer side to 224 pixels while maintaining the aspect ratio, and padding the remaining pixels with zeros.
> > >
> > > The retrieval performances are as follows:
> > >
> > > | Method | Resolution | Input Strategy | Rank-1 | Rank-5 | Rank-10 | mAP   |
> > > | - | - | - | - | - | - | - |
> > > | FAFA   | 192 × 384  | Resized Positional Embeddings | 44.21  | 64.97  | 71.76   | 53.89 |
> > > | FAFA   | 224 × 224  | Directly Resizing Images| 45.62  | 65.58  | 72.43   | 54.63 |
> > > | FAFA   | 224 × 224  | Ours (Resizing + Padding) | 46.54  | 66.21  | 73.12   | 55.60 |
> > >
> > > We can observe that although the largest input resolution (192 × 384) incurs the greatest computational cost, it yields the worst retrieval performance, which is most likely due to the deterioration of feature extraction caused by resizing the positional embeddings. Directly resizing images to 224 × 224 disrupts the correct aspect ratios and has a negative impact on the feature extraction. In contrast, our final choice effectively avoids these issues and thus achieves the best performance.
> > >
> > > [1] Eva-02: A visual representation for neon genesis[J]. Image and Vision Computing, 2024.
> > >
> > > We warmly welcome continuing to clarify your follow-up questions. And it would be even better if you could consider raising the score. Haha.

---

> > > > ### Comment · Reviewer_AeXC · 2025-08-05
> > > >
> > > > I appreciate the authors' additional experimental results and discussions. All of the major concerns I raised have been addressed during the rebuttal period.

---

### Official Review · Reviewer_ZBdY · 2025-06-30

**Clarity:** 3
**Significance:** 3
**Originality:** 3
**Rating:** 4
**Confidence:** 5

**Summary:**

This paper introduces a novel Composed Person Retrieval (CPR) task, which leverages both a reference image and a textual relative caption to retrieve a target person image from a large-scale gallery. To support this task, the authors propose a scalable pipeline for automatic triplet synthesis, utilizing large language models (LLMs) and a fine-tuned generative model. Based on this pipeline, they construct SynCPR, a fully synthetic CPR dataset at million-scale. Furthermore, they present a Fine-grained Adaptive Feature Alignment (FAFA) framework tailored for the CPR task. To enable objective evaluation on real-world data, the authors also annotate existing clothes-changing person retrieval datasets with CPR-specific labels.

**Questions:**

1. Regarding the triplet synthesis pipeline, the authors fine-tune DiT using different LoRA strength values β to introduce style diversity, but cause the images in SynCPR exhibit varying degrees of realism depending on β. However, it seems to lack experiments to verify whether using multiple β values leads to better performance compared to a single fixed β and examine whether the inclusion of less realistic images has any negative impact on model performance. Could you clarify this point？

2. Among the four settings used for comparison with existing methods, the pretraining data varies across different settings, and some baselines were not trained for the text+image (T+I) task—particularly in the first two settings. Under such circumstances, can these comparisons reliably support the claimed superiority of the FAFA approach?

3. In the FAFA framework, why are the fine-grained features derived from the target image rather than the reference image?

**Ethical Concerns:**

["NO or VERY MINOR ethics concerns only"]

**Final Justification:**

Most of my concerns have been addressed by the authors, I will maintain the initial score.

**Limitations:**

1. CPR appears to require both text and image queries simultaneously, which may limit its applicability in scenarios where only one modality is available.

2. ITCPR focuses mainly on clothing change, thus leading to limited evaluation of other potential variations.

**Paper Formatting Concerns:**

There is no issue with the formatting.

**Quality:**

3

**Strengths And Weaknesses:**

Strengths:
1. Quality: ① The pipeline for constructing SynCPR is technically sound, with identity consistency achieved via innovative single-image pair synthesis and realistic images generated by a model fine-tuned specifically for this task. ② The experiments are comprehensive, including extensive ablation studies and comparisons with existing methods across multiple settings. ③ ITCPR provides rigorous evaluation on real clothes-changing datasets with false-negative mitigation, making the assessment more convincing.
2. Clarity: The paper demonstrates clear writing and a coherent structure, with a fairly comprehensive description of the experimental implementation details.
3. Significance: ① CPR complements existing retrieval tasks by combining text and image queries, demonstrating promising applicability in real-world scenarios. ② This work introduces two novel datasets: SynCPR, which eliminates the need for costly manual annotation, and ITCPR, which is derived from existing clothes-changing person retrieval datasets with additional annotations.
4. Originality: ① First formalization of CPR, bridging IPR/TPR/CIR gaps. ② A fine-tuned generative model for images that closely resemble real-world persons generation.

Weaknesses:
1. Quality: Experiments focus on clothing changes (ITCPR) while performance on other variations (e.g., scene) is unexplored
2. Clarity: The  icons in some figures, such as Figure 1 (b) and 3 are not distinguishable.
3. Significance: ITCPR primarily focuses on clothing changes and lacks coverage of other types of variations.
4. Originality: The distinction between CPR and CIR could be articulated more clearly.

---

> ### Author Rebuttal · Authors · 2025-07-30
>
> We sincerely thank you for your efforts and time, as well as your overall positive evaluation. In response to your crucial comments, we provide detailed clarifications below.
>
> ## Responses to Weaknesses:
>
> **A1.1: Focus Mainly on Appearance Variations**
>
> When manually annotating the ITCPR test set, we primarily focus on appearance changes. This focus reflects many practical scenarios. For instance, when looking for a missing person, we may have a previous photograph in which the appearance often differs significantly from that on the day of disappearance. Considering that other types of variations, as you pointed out, such as changes in scenes or hairstyles and so on, also hold additional practical value, we plan to add more annotations to align with our SynCPR training set, which includes various types of variations.
>
> **A1.2: Figure Icon Issues**
>
> Thank you for pointing out the issue with the icons in Figures 1(b) and 3. In the revised manuscript, we will refine both figures by adding clearer legends and supplementary annotations to improve readability and ensure that all icons are easily distinguishable.
>
> **A1.3:** This concern overlaps with our response provided in **A1.1**. Please refer to A1.1.
>
> **A1.4: Distinction Between CPR and CIR Tasks**
>
> Composed Image Retrieval (CIR) aims to retrieve a target image based on a query composed of a reference image and a relative caption that describes the difference between the two images [1]. Composed Person Retrieval (CPR) aims to retrieve a target person image. Therefore, the main distinction between CPR and CIR lies in the granularity and strictness of retrieval constraints. CIR focuses on retrieving images matching the semantic modifications described by a combination of textual and visual queries, and there are no strong constraints between the reference and target images. Typical CIR examples are provided by Figure 8 in the arXiv version of reference [1]. In contrast, CPR requires the retrieved target person to share the SAME ID as the reference individual, supplemented with additional details such as clothing attributes, thereby imposing stronger constraints. Moreover, since all images in CPR are of persons, the retrieval model must precisely distinguish subtle semantic differences among numerous visually similar candidates. Therefore, compared with CIR, the CPR task requires stronger capabilities in image relevance assessment and fine-grained semantic differentiation.
>
> [1] Zero-Shot Composed Image Retrieval with Textual Inversion. ICCV2023 or arXiv:2303.15247
>
> ## Responses to Questions:
>
> **A2.1: Clarification on Using Multiple β Values**
>
> When the data distribution of the test domain is unknown, increasing the diversity of training data helps to enhance the model's adaptability under various distributions. To this end, via LLM-generated texts, our method first ensures diverse textual descriptions. Then, the image style diversity is controlled by adjusting the β value. As shown in supplementary Figure S11, different β values correspond to different image styles. For example, β=0 corresponds to purely generated (artificial) style images; β≈0.3 corresponds to photographic-quality images; β=1 corresponds closely to surveillance-style images. Therefore, to achieve diverse image styles and maintain a strong sense of realism for the majority of images, we adopt a strategy of setting β=1 with 50% probability during inference, while randomly sampling β values in other cases to ensure broad distribution coverage.
>
> To verify the impact of different β strategies on the retrieval performance of FAFA, we first randomly select 10,000 triplets (5,000 quadruples) from the SynCPR dataset as a benchmark training set. Then, we respectively generate different datasets using fixed β=1, β=0, and fully random β. To ensure a fair comparison, the images that do not meet the filtering criteria are regenerated. The performance comparison is as follows:
>
> | Method | Train Data | Rank-1 | Rank-5 | Rank-10 | mAP   |
> | - | - | - | - | - | - |
> | FAFA   | β=0 | 33.70  | 56.00  | 62.81   | 43.56 |
> | FAFA   | β=1| 40.69  | 61.53  | 69.57   | 51.35 |
> | FAFA   | Random β | 40.51  | 61.58  | 69.26   | 51.27 |
> | FAFA   | Ours (mixed β) | 41.51  | 62.40  | 69.71   | 52.03 |
>
> The results clearly show that predominantly using unrealistic images (β=0) seriously harms retrieval performance. Although a fully random β can provide broad distribution coverage, a high proportion of less realistic images slightly reduces retrieval effectiveness. Conversely, relying exclusively on highly realistic images (β=1) narrows the data distribution, causing a slight performance drop due to varying image styles from multiple sources within ITCPR. Finally, the strategy of combining realistic images with broad distributional coverage achieves the best performance. We will add the above analysis to the revised manuscript.
>
> **A2.2: Reliability of FAFA Superiority Claims in Table 1**
>
> In Table 1, we aim to compare all promising solutions for the CPR task. To objectively evaluate the performance of each method on the CPR task, the selected training datasets for these methods can be either the ones recommended in their original works or the widely recognized datasets within their respective fields. We observe that except for the first two experimental settings (which are not specifically trained for text+image retrieval) failing to reliably demonstrate FAFA's superiority, the comprehensive results under all other settings effectively validate the superior performance of FAFA.
>
> It should be noted that the first two settings are actually essential for demonstrating the validity of the ITCPR dataset construction and related experimental protocols. In other words, using only text, only images, or simply combining the two leads to poor performance. Only by effectively extracting complementary information from combined queries, as demonstrated by FAFA, can higher retrieval accuracy be achieved.
>
> **A2.3: Fine-grained Features Derived from Target Image**
> The reasons for extracting fine-grained features solely from the target image are as follows:
>
> 1. **Retrieval efficiency**: If fine-grained features are used on both query and target sides, matching will require solving a (k × k) bipartite graph, significantly increasing computational complexity and complicating loss design. Additionally, this approach violates our "dynamic alignment" principle, which emphasizes "selection on one side and aggregation on the other."
>
> 2. **Clear role differentiation aligned with CPR characteristics**:
>
>    * Query side: Combines a reference image and relative textual description, inherently requiring information integration to predict "how the appearance should change."
>    * Target side: Must identify the correct target from numerous candidates, inherently requiring preservation of local discriminative features for fine-grained differentiation.
>
> 3. **Avoiding multimodal noise interference**: On the query side, the image and textual description are integrated through the Q-Former, resulting in a semantically-completed holistic representation. The Masked Feature Reasoning (MFR) further reinforces this integration. If this unified representation is decomposed back into patch-level fine-grained vectors, textual noise and semantic confusion will be introduced at the patch level, negatively affecting retrieval performance.
>
> ## Responses to Limitations:
>
> **A3.1: Modality Deficiency Experiments**
>
> Thank you for providing such an insightful suggestion. Our FAFA can handle the situation of missing modalities by making minor adjustments to the training strategy. Specifically, while keeping all other settings unchanged, we retrain FAFA on SynCPR data with several modality-deficient scenarios: 70% complete inputs, 15% missing textual inputs (blank text), and 15% missing images (purely black images). During inference, the same modality-deficiency settings are applied. The resulting test performances are as follows:
>
> | Method  | Test Data  | Rank-1 | Rank-5 | Rank-10 | mAP   |
> | - | - | - | - | - | - |
> | FAFA    | Complete   | 45.87  | 65.53  | 72.52   | 54.90 |
> | SOLIDER | Image-only | 8.45   | 18.48  | 23.89   | 13.74 |
> | FIRe2   | Image-only | 10.76  | 22.84  | 29.29   | 17.00 |
> | FAFA    | Image-only | 12.08  | 24.43  | 33.65   | 19.38 |
> | RDE     | Text-only  | 26.43  | 47.41  | 56.45   | 36.43 |
> | RaSa    | Text-only  | 28.02  | 49.23  | 57.77   | 38.04 |
> | FAFA    | Text-only  | 29.61  | 51.59  | 60.22   | 39.84 |
>
> Comparing these results to those reported in Table 1, we can see that the modified FAFA slightly reduces performance under combined inputs but remains highly effective. Moreover, FAFA performance under modality-deficient conditions surpasses many dedicated image-based person retrieval (IPR) [2,3] and text-based person retrieval (TPR) methods [4,5], which are specifically trained on their respective benchmark datasets.
>
> [2] Beyond appearance: a semantic controllable self-supervised learning framework for human-centric visual tasks. CVPR, 2023
>
> [3] Exploring fine-grained representation and recomposition for cloth-changing person re-identification. TIFS, 2024
>
> [4] Noisy-correspondence learning for text-to-image person re-identification. CVPR, 2024
>
> [5] RaSa: relation and sensitivity aware representation learning for text-based person search. IJCAI, 2023
>
> **A3.2:** Please see the response to A1.1 above.

---

> > ### Comment · Reviewer_ZBdY · 2025-08-05
> >
> > Thank you for your response. Most of my concerns have been addressed by the authors.

---

### Official Review · Reviewer_iMvu · 2025-07-01

**Clarity:** 3
**Significance:** 2
**Originality:** 2
**Rating:** 4
**Confidence:** 4

**Summary:**

The paper proposes a novel task, Composed Person Retrieval, which integrates visual and textual information to enhance person retrieval accuracy.

**Questions:**

- Could the authors provide a detailed analysis of the computational complexity and resource requirements for the triplet synthesis pipeline, particularly regarding LLM and diffusion model inference times, to assess its practicality for large-scale use?
- Can the authors clarify how the FAFA framework handles extreme cases, such as significant occlusions or low-quality reference images, and provide corresponding experimental results to demonstrate robustness?
- Could the authors discuss the potential for extending the CPR task to include multiple reference images or textual descriptions, as mentioned in the conclusion, and provide preliminary results or insights?

**Ethical Concerns:**

["NO or VERY MINOR ethics concerns only"]

**Final Justification:**

Thanks for your efforts in addressing my concerns. I believe the issues have been mostly well explained, and I will maintain my positive score.

**Limitations:**

Please find the above section.

**Paper Formatting Concerns:**

Thanks for your efforts in addressing my concerns. I believe the issues have been mostly well explained, and I will maintain my positive score.

**Quality:**

3

**Strengths And Weaknesses:**

- One weakness is the heavy reliance on the synthetic SynCPR dataset for training. While the generation process is impressive, there is no in-depth analysis of the domain gap between the synthetic training data and the real-world test data (ITCPR).
- CPR task address the practical need to combine visual and textual queries for person retrieval, which is underexplored compared to image-based or text-based retrieval.
- The paper is well-written and clearly structured, and The figures are effective at conveying the core ideas. The experiments and results are presented with clarity.
- The proposed automatic data synthesis pipeline is scalable and innovative, generating a large-scale synthetic dataset to mitigate the lack of annotated data.
- The originality of the FAFA framework is somewhat tempered by its reliance on existing architectures (e.g., BLIP-2), with incremental improvements over previous CIR methods.
- Manually annotated ITCPR test set, containing 2,225 triplets, is relatively small, which might not be sufficient to draw robust conclusions about the generalization capabilities of the proposed methods.

---

> ### Author Rebuttal · Authors · 2025-07-30
>
> We appreciate your efforts and time, as well as your overall positive assessment. In response to your crucial concerns, detailed clarifications are provided below.
>
> ## Response to Weaknesses:
>
> **A1.1: In-depth analysis of the domain gap between synthetic training data and real-world test data**
>
> Thank you for this insightful suggestion. For the Composed Person Retrieval (CPR) task, the textual data primarily describes relative appearance differences between the target and reference person images. Therefore, the main source of potential domain discrepancy originates from differences in visual distributions between synthetic and real-world images.
>
> During the construction of the SynCPR dataset, apart from knowing that the test images are real images, we have no prior access to any distributional information about the real-world test images. Consequently, we aim to ensure that the majority of images in the SynCPR are highly realistic and sufficiently diverse in distribution. Specifically, the image generator used in the SynCPR dataset is fine-tuned on the real-world surveillance dataset CUHK-PEDES. During inference, we adopt a mixed strategy: setting β=1 with a 50% probability to ensure highly realistic images, and randomly setting β with a 50% probability to encourage broad distributional diversity (partially overlapping with the real-world test domain). Its purpose is to minimize distributional differences in the absence of explicit test-domain information.
>
> To quantitatively assess this strategy, we calculate the Fréchet Inception Distance (FID) using different settings. Specifically, we randomly sample 5,000 image groups (10,000 images in total) from SynCPR and compute the FID against all images annotated in the real-world ITCPR test set, using the same prompts but different β settings. The results are as follows:
>
> | Strategy | β=0    | β=1   | Ours (mixed) |
> | -------- | ------ | ----- | ------------ |
> | **FID**  | 134.85 | 96.69 | **89.24**    |
>
> These results demonstrate that our current strategy effectively minimizes the domain gap between synthetic training images and real-world test images.
>
> **Theoretical justification**: Generally, when two distributions are narrow with minimal overlap, the FID value is relatively large. Conversely, when one distribution is broad (representing diverse SynCPR data) and the other is narrow (representing ITCPR data), with some overlap, the FID tends to be lower [1, 2].
>
> [1] GANs Trained by a Two Time-Scale Update Rule Converge to a Local Nash Equilibrium. NIPS, 2017
> [2] Training Generative Adversarial Networks with Limited Data. NIPS, 2020
>
> **A1.5: Improvements of retrieval framework**
>
> In the field of person retrieval, especially for the text-to-image retrieval task, it is a common practice to further fine-tune pre-trained Vision-Language Pretraining (VLP) models. Leveraging pre-trained knowledge enables rapid and effective alignment between textual descriptions and visual content, thereby achieving superior performance. Given that the CPR task also inherently involves cross-modal retrieval, we therefore select the visual encoder and Q-Former components of BLIP-2 as the foundational architecture. Importantly, we further propose three new modules to specifically address the fine-grained nature of person retrieval: Fine-grained Dynamic Alignment (FDA), adaptive Feature Diversity (FD) supervision, and bidirectional Masked Feature Reasoning (MFR), aiming to make full use of complementary information between reference images and relative captions. Comprehensive experiments and ablation studies verify the positive effect of each component on performance improvement. Although these improvements only constitute one aspect of our overall contributions, they have shown indispensable value in the context of our proposed retrieval framework.
>
> **A1.6: Scale of test dataset**
>
> In the Text-based Person Retrieval (TPR) field, there are two widely-used benchmark datasets, namely the CUHK-PEDES and the RSTPReid. The former contains 3,074 images and 6,156 texts in its test set, while the latter only includes 1,000 images and 2,000 texts for testing. In contrast, our ITCPR test set contains 2,225 triplets (reference image, relative caption, target image) and requires more complex and meticulous annotations; thus, its scale is comparable to that of similar datasets and is capable of supporting the validation of model effectiveness. Considering that this is the first dataset in the field of CPR, the ITCPR can be expanded in terms of both scale and quality. Hence, we will follow your comment to continue with the annotation and expansion work in our subsequent steps while retaining the current version (so that other methods can conduct fair comparisons).
>
>
> ## Responses to Questions:
>
> **A2.1 Computational Efficiency of Data Synthesis**
>
> We conduct efficiency evaluations on a single H800 GPU following the configurations described in Section 4.1. The results show that: (1) With batch sizes set to 8, the average inference time for generating a single textual quadruple using Qwen2.5-72B is approximately 6.45 seconds. (2) The generation of a pair of person images takes around 3.23 seconds. (3) Filtering a generated image pair using Qwen2.5-VL-32B requires about 0.38 seconds. Therefore, synthesizing one annotated quadruplet, which consists of two triplet annotations, takes an average of 10.06 seconds, with each individual triplet annotation requiring 5.03 seconds. Consequently, approximately 17,000 annotated triplets can be generated daily on a single GPU, which is approximately 24 times faster than manual annotation (it takes an average of 120 seconds for one triplet), making our pipeline highly practical for large-scale applications. It is worth mentioning that this synthesis process only needs to be carried out once and does not require repeated computation. Furthermore, the SynCPR dataset will certainly be open-sourced, thus avoiding the need for redundant generation efforts in the future.
>
> **A2.2 Robustness of FAFA Framework**
>
> As detailed in Appendix C.1, we incorporate strong data augmentation techniques during FAFA training, including random horizontal flipping, random cropping and padding, and random erasing. These augmentations enhance the robustness of the model to severe occlusions and low-quality reference images. Specifically, we evaluate the FAFA model trained on the SynCPR dataset under various perturbations applied to the ITCPR dataset. These perturbations include random erasing of image regions (covering 5%-30% of the image area) to simulate severe occlusions, and applying Gaussian blurring with randomly chosen standard deviations (with σ from 0.1 to 3.0) to simulate low-quality images. The results are summarized below:
>
> | Method | Test Data | Rank-1 | Rank-5 | Rank-10 | mAP   |
> | - | - | - | - | - | - |
> | FAFA   | ITCPR | 46.54  | 66.21  | 73.12   | 55.60 |
> | FAFA   | + Gaussian Blur  | 46.19  | 65.62  | 72.66   | 55.12 |
> | FAFA   | + Random Erasing | 45.00  | 64.31  | 71.84   | 54.07 |
> | FAFA   | + Erasing + Blur | 44.10  | 64.08  | 71.48   | 53.39 |
>
> The results indicate that our FAFA demonstrates strong robustness. It can maintain its effectiveness even under severe occlusion and low-quality conditions, and the performance degradation is within a controllable range. Additionally, in response to Reviewer ZBdY's comment A3.1, we further discuss the performance of FAFA when encountering modality deficiencies, and the favorable results also support the robustness of our method.
>
> **A2.3 Feasibility of Multi-image and Multi-text Input**
>
> Thank you for keenly identifying this promising direction for future research, as we envisioned in the conclusion. However, it is difficult to provide quantitative results in a short time because the corresponding test sets require high-quality manual annotations, and there is currently a lack of publicly available datasets that support multiple reference images and textual descriptions.
>
> Nevertheless, from the perspectives of dataset synthesis and methodological extensions, we have designed a feasible technical solution. Firstly, for data generation, the established SynCPR dataset can be used as training data for image generators, enabling the training of consistent generation or image-editing models [3,4]. Such methods are helpful for generating multiple diverse images of the same individual based on one reference image combined with varied textual descriptions. Then, the textual generation and filtering pipeline proposed in our paper can naturally extend to construct a dataset supporting multiple references for the CPR task.
>
> Regarding the training of the retrieval model, simple modifications can enable it to adapt to the situation of multiple images or texts. One straightforward approach is to increase the dimensionality of image input and use placeholders when necessary. Meanwhile, multiple textual descriptions can be naturally concatenated into a single, extended query sentence.
>
> [3] Ominicontrol: Minimal and universal control for diffusion transformer. ICCV, 2025
>
> [4] In-Context Edit: Enabling Instructional Image Editing with In-Context Generation in Large Scale Diffusion Transformer. arXiv:2504.20690

---

> ### Comment · Area_Chair_hraN · 2025-08-05
>
> Reviewer iMvu, please engage in the discussion period. I understand that the review period started over a weekend, but we only have a few days remaining in the (now slightly extended) discussion period. The authors have provided a thoughtful response to your review and you are obligated to respond to it. You should share with the authors if they addressed questions or concerns you had, and seek clarification about any questions or concerns that remain. Please post your response as soon as you can so that there is time for the authors to follow up and discussion to progress as needed.

---

> ### Comment · Reviewer_iMvu · 2025-08-05
>
> Thanks for your efforts in addressing my concerns. I believe the issues have been mostly well explained, and I will maintain my positive score.

---

### Note · Authors · 2025-08-12

Thank you to the PC, AC, and all the reviewers for your hard work and rigorous discussions.

We are very pleased to have clarified all the major concerns of the four reviewers and have verified the effectiveness of the proposed method across different aspects, including the synthetic data strategy, domain gap, image input strategy, storage and retrieval efficiency, occlusion, blur, and single-modality usage, through additional detailed experiments.

Moreover, in response to the positive suggestions from the reviewers, such as adding examples of application scenarios and the selection strategy for fine-tuning datasets, we have incorporated them into the revised version, and we will include all other valuable supplementary experimental results in the appendix. In terms of ethical risks, we have always adhered to the concept of "Tech for Good." In the revised version, we present the technical process more transparently and detail concrete measures to avoid potential risks and misuse. Building on Appendix Fig. S10, we add more intuitive visualizations to make the coverage of the synthetic data across different ages, genders, and ethnicities clearer. We also place greater emphasis on quantitative metrics and data security, for example by adding demographic group statistics, reporting fairness metrics (Demographic Parity and Equalized Odds), and adding digital watermarks to the datasets. We ensure that the technology in this paper is limited to the academic field and can provide a new and beneficial idea for researchers in the same field.

Finally, all four reviewers assigned positive scores of at least 4 and fully affirmed the novelty and contributions of the newly proposed Composed Person Retrieval (CPR) based on synthetic images, in terms of both methodology and datasets, including the first data synthesis method, the first CPR test dataset, and the increments compared with existing methods.

Thank you again to the PC, AC, and all the reviewers for your hard work.

---

### Decision · Program_Chairs · 2025-09-17

**Decision:**

Accept (poster)

**Comment:**

The paper introduces a new task, Composed Person Retrieval (CPR), which uses both a reference image and a textual description to identify a target person from a large gallery. Its core contributions include a scalable synthetic data generation pipeline, a fine-grained retrieval framework, and a manually annotated benchmark for evaluation.

Reviewers thought the paper proposed a novel and relevant retrieval task, presenting a technically sound synthetic data generation pipeline, and introducing a well-structured retrieval framework. Reviewers also appreciated the contributed dataset, and noted the comprehensiveness of the experiments and ablations. The clarity of writing and the potential impact of CPR in real-world applications were also seen as strengths by the general reviewers (I’ll discuss more about the ethics reviews below).

Initial concerns from reviewers included the scale and diversity of the test set, the domain gap between synthetic and real data, justification of FAFA components, computational cost, and practical utility of the CPR task. However, these were largely resolved during the rebuttal and discussion, which the reviewers indicated sufficiently addressed their concerns.

With this said, this paper was flagged for ethics reviews due to the human data and surveillance applications. The ethics reviewers raised valid concerns regarding fairness, bias in synthetic data generated using LLMs, and the dual-use potential of CPR for surveillance. Specifically, they pointed out the need for more transparency about demographic representation, fairness across groups, and the societal risks of surveillance misuse. In response, the authors made several commitments for the revised version including incorporating demographic diversity statistics and fairness metrics, adding discussion of bias mitigation strategies like prompt design and filtering, potentially embedding digital watermarks in datasets for traceability, and limiting dataset release to the academic community under responsible-use agreements. These are meaningful steps. It’s not clear that all of them are feasible for the camera ready submission, but the authors should perform due diligence to address the ethics reviewer concerns as best as possible in the camera ready. The paper should have an explicit and dedicated ethics/limitation section that states clearly what steps were taken to mitigate the ethics reviewers very valid concerns.

In addition to the ethics-focused revisions, the authors are encouraged to include the additional visualizations, extended application discussions, and clarifications they promised during the rebuttal (e.g., on FD/MFR intuitions and visualizations, and the updated figures) in the camera ready submission.

Overall, this is a strong and comprehensive paper that introduces a novel task with practical significance, and the authors have effectively addressed reviewer concerns. The AC recommends acceptance.